# Characterization of aerosol growth events over Ellesmere Island during summers of 2015 and 2016

Samantha Tremblay[1], Jean-Christophe Picard [1], Jill O. Bachelder[1], Erik Lutsch[2], Kimberly Strong[2], Pierre Fogal[2], W. Richard Leaitch[3], Sangeeta Sharma[3], Felicia Kolonjari[3], Christopher J. Cox[4], Rachel Y.-W. Chang[5], Patrick L. Hayes[1]

[1]Department of Chemistry, Université de Montréal, Montréal, Québec, Canada
[2]Department of Physics, University of Toronto, Toronto, Ontario, Canada
[3]Climate Research Division, Environment and Climate Change Canada, Toronto, Ontario, Canada
[4]Cooperative Institute for Research in Environmental Sciences (CIRES), Boulder, CO, USA and NOAA Physical Sciences Division, Boulder, CO, USA
[5]Department of Physics and Atmospheric Science, Dalhousie University, Halifax, Nova Scotia, Canada

*Correspondence to*: Patrick L. Hayes (patrick.hayes@umontreal.ca), Rachel Chang (rachel.chang@dal.ca)

**Abstract.** The occurrence of frequent aerosol nucleation and growth events in the Arctic during summertime may impact the region's climate through increasing the number of cloud condensation nuclei in the Arctic atmosphere. Measurements of aerosol size distributions and aerosol composition were taken during the summers of 2015 and 2016 at Eureka and Alert on Ellesmere Island in Nunavut, Canada. These results provide a better understanding of the frequency and spatial extent of elevated Aitken mode aerosol concentrations as well as of the composition and sources of aerosol mass during particle growth. Frequent appearance of small particles followed by growth occurred throughout the summer. These particle growth events were observed beginning in June with the melting of the sea ice rather than with polar sunrise, which strongly suggests that influence from the marine boundary layer was the primary cause of the events. Correlated particle growth events at the two sites, separated by 480 km, indicate conditions existing over large scales play a key role in determining the timing and the characteristics of the events.

In addition, aerosol mass spectrometry measurements were used to analyze the size-resolved chemical composition of aerosols during two selected growth events. It was found that particles with diameters between 50 and 80 nm (physical diameter) during these growth events were predominately organic with only a small sulphate contribution. The oxidation of the organics also changed with particle size, with the fraction of organic acids increasing with diameter from 80 to 400 nm.

The growth events at Eureka were observed most often when the temperature inversion between the sea and the measurement site (at 610 m ASL) was non-existent or weak, presumably creating conditions with low aerosol condensation sink and allowing fresh marine emissions to be mixed upward to the observatory's altitude. While the nature of the gaseous precursors responsible for the growth events are still poorly understood, oxidation of dimethyl sulphide alone to produce particle phase sulphate or methanesulphonic acid was inconsistent with the measured aerosol composition, suggesting the importance of other gas phase organic compounds condensing for particle growth.

## 1 Introduction

Surface aerosol concentrations in the Arctic are characterized by a distinct seasonal cycle, with high mass loadings in the winter followed by very low mass loadings in the summer (Sharma et al. 2004; Quinn et al. 2007; Engvall et al. 2008; Sharma et al. 2013; Tunved et al. 2013; Croft et al. 2016a; Nguyen et al. 2016). This cycle is caused by different transport patterns and by changes in wet deposition, with wintertime air influenced by pollution originating from continental regions at lower latitudes such as Europe, Siberia and even South Asia (Stohl 2006). In contrast, during summertime, air masses originating from lower latitudes experience greater wet deposition during transport northwards, resulting in very few particles arriving to the north. Consequently, local sources dominate the surface aerosol. In wintertime, Arctic air near the surface spends about one week continuously above 80°N, whereas in summertime the air near the surface spends about two weeks continuously above 80°N (Stohl 2006), also increasing the relative importance of aerosols originating in the Arctic. The nature and sources of aerosols of Arctic origin during summertime are still poorly understood, although marine and snow or ice-related sources have been suggested in the past (Leck and Bigg 2005b; Fu et al. 2013; Willis et al. 2016). As the Arctic continues to warm and summer sea-ice coverage decreases, contributions from marine sources will likely increase while snow or ice-related sources will decrease. In addition, increased shipping and industrial activities during the Arctic summer in the future could completely shift the relative importance of natural and anthropogenic aerosol sources (Croft et al. 2016a).

In tropical marine locations, new particle formation tends to occur in the upper part of the troposphere, usually at the outflow of clouds, and these particles are entrained to the surface through mixing, which contributes to relatively stable aerosol size distributions (Hoppel et al. 1986; Clarke et al. 2006). In contrast, modelling studies of the Arctic summer show that persistent cloud and drizzle causes wet deposition and results in low condensation sinks at the surface (Browse et al. 2014; Croft et al. 2016a). These same studies show that these conditions can favour particle nucleation followed by growth between drizzle events. This is supported by surface observations of aerosol size distributions in the Arctic at Alert and Ny-Ålesund that show an annual cycle during which summertime surface aerosols exhibit much smaller particle diameter than wintertime aerosols (Tunved et al. 2013; Croft et al. 2016a). Additional surface observations have suggested that new particle formation could be the source of these small particles, with dimethyl sulphide (DMS) emitted from the ocean being a key gaseous precursor of less volatile species, such as sulphuric acid and methanesulphonic acid, that contributes to aerosol mass (Asmi et al. 2011; Chang et al. 2011; Karl et al. 2011; Karl et al. 2012; Leaitch et al. 2013).

Sulphuric acid has long been known to contribute to new particle formation and growth events (Twomey 1977; Charlson et al. 1992; Napari et al. 2002; Lohmann and Feichter 2005; Kirkby et al. 2011; Almeida et al. 2013; Croft et al. 2016b). More recent work has shown that in coastal Arctic environments, ammonia from sea-bird colonies can contribute to new particle formation (Croft et al. 2016b). These findings are further supported by previous measurements of aerosol composition using a volatility tandem differential mobility analyzer system installed near Ny-Ålesund, Svalbard (Giamarelou et al. 2016) suggested that 12 nm particles were predominately ammonium sulphate, although it was not

possible in that study to conclusively distinguish ammonium sulphate from organics with similar volatility. In addition, organic compounds, especially those with lower volatilities, have also been found to contribute secondary aerosol mass to particle growth and nucleate new particles in forested and anthropogenically influenced regions (Allen et al. 2000; Zhang et al. 2009; Pierce et al. 2012; Riipinen et al. 2012) as well as in laboratory studies (Kirkby et al. 2016; Trostl et al. 2016). Box

models have inferred the contribution of non-sulphur species (i.e. organic compounds) to aerosol growth in Greenland (Ziemba et al. 2010) and in tropical marine cloud outflow (Clarke et al. 1998). Burkart et al. (2017) provided indirect evidence that organic compounds contribute to aerosol growth in high-latitude marine environments using both microphysical modeling of a particle growth event as well as cloud condensation nuclei (CCN) hygroscopicity measurements. In a comparison of ship-borne observations in the Canadian Arctic in 2014 and 2016, Collins et al. (2017)

found that increased activity in marine microbial communities along with greater solar radiation and lower sea ice concentrations contributed to new particle formation and growth. Recent work by Mungall et al. (2017)  in the Canadian Arctic also suggests that a photo-mediated marine source of oxygenated volatile organic compounds could produce precursor vapors for new particle formation or growth. Furthermore, iodine may be important for particle nucleation in the Arctic (Mahajan et al. 2010; Allan et al. 2015; Sipila et al. 2016; Raso et al. 2017), although the processes leading to either

nucleation or particle growth are not necessarily the same.

       The GEOS-Chem chemical transport model has been used to model particle formation and size distributions in the Arctic (Bey et al. 2001; Wild and Prather 2006; Croft et al. 2016a; Croft et al. 2016b; Christian et al. 2017). Recent work using GEOS-Chem with the size-resolved aerosol microphysics package TOMAS (Croft et al. 2016a; Croft et al. 2016b) analyzed size distributions of aerosols measured in the Arctic, and showed that GEOS-Chem-TOMAS underestimates

Aitken mode particle sizes during the summertime. It was also shown that new particle formation can be driven by neutralization reactions, where missing ammonia emissions can be accounted for by seabird colonies. However, this work acknowledged poor constraints on marine primary aerosol and secondary organic aerosol precursors. These results demonstrate the difficulties that the GEOS-Chem model has in predicting particle size distributions for the Aitken mode during summertime, which is presumably due to missing processes contributing to particle growth (e.g. the condensation of

semi-volatile or low-volatility vapors). Similar discrepancies are also observed in the chemical transport model GLOMAP (Global Model of Aerosol Processes) (Korhonen et al. 2008; Browse et al. 2014), with a low bias observed for either Aitken or Accumulation mode aerosols.

       In this study we present direct measurements of size-resolved aerosol chemical composition using mass spectrometry to better understand the processes contributing to aerosol growth during the summertime in the Canadian High

Arctic. These measurements, as well as those of aerosol number size distribution, were conducted at Eureka, Nunavut on Ellesmere Island in the Canadian Arctic Archipelago. For comparison, aerosol size distributions measured at Alert, which is located further north on Ellesmere Island, are also reported. Numerous concomitant events in which small particles appear and then grow are observed at both sites throughout the summer, resulting in large variations in the number concentration of particles with diameters smaller than 100 nm. The mass spectrometry measurements indicate that these ultrafine particles

(<100 nm) were predominately organic during the observed growth events. This work builds on other studies that have indirectly characterized the organic content of Aitken mode aerosols in the Arctic (Burkart et al. 2017) and have measured oxidized volatile organic compounds in the Arctic atmosphere (Mungall et al. 2017). Taken together, these results provide important evidence that the condensation of lower volatility organic vapors on particle surfaces may be responsible, at least in part, for the particle growth events that are frequently measured at two sites on Ellesmere Island (e.g. approximately 20 events during summer 2016 at Eureka).

## 2 Experimental

### 2.1 Field Site Information and Aerosol Sizing Instrumentation

The primary measurement site for surface aerosols was the Polar Environment Atmospheric Research Laboratory (PEARL) (Fogal et al. 2013) located on Ellesmere Island in Nunavut, Canada (80.05° N, 86.42° W). The PEARL Ridge Laboratory (RidgeLab) is located 610 m above sea level and 11 km northeast of the Environment and Climate Change Canada (ECCC) Eureka Weather Station, located at sea level. Radiosondes are launched twice a day from the Weather Station at 00:00 UTC and 12:00 UTC and are used in this work to evaluate the vertical temperature profile and presence of temperature inversions between sea level and the altitude of the RidgeLab. Solar radiation data were measured by a pyranometer (Kipp & Zonen CM 21) at the Surface and Atmospheric Flux, Irradiance and Radiation Extension (SAFIRE) site, situated near Eureka Weather Station at 85 meters above sea level (79.98° N, 85.93° W). A map showing the PEARL RidgeLab, Eureka weather station and SAFIRE is provided in Figure S2.

A scanning mobility particle sizer (SMPS, TSI 3034) measured the aerosol size distribution for diameters between 10 and 487 nm in 54 channels, while an optical particle counter (OPC, Met One GT-526S) measured the aerosol size distribution at diameters between 0.3 and 10 µm in six channels. Both instruments were connected to a common inlet, which is described in greater detail in the SI. Following the work of DeCarlo et al. (2004), the mobility diameter measured by the SMPS was assumed to be equal to the physical diameter, which would be valid if the sampled particles were spherical and contained no voids. This is a reasonable assumption given the secondary origin of the observed particles. It was further assumed that the OPC diameter was equal to the physical diameter, given that the Mie scattering curve of the ambient aerosols was likely within 10% of that of the calibration particles composed of polystyrene latex spheres.

Measurements from these instruments are reported for a period starting in July 2015 through September 2016, and thus consist of one full 2016 summer season and the full summer month of August 2015. (Wintertime measurements were taken too, but are not presented in this article.) Both the OPC and SMPS data were recorded every three minutes, and then averaged hourly for analysis and comparison to other data sets. Agreement between the SMPS and OPC was evaluated by comparing the particle number concentration between 300 – 487 nm measured by the SMPS against the concentration measured by the OPC for approximately the same range of particle diameters (300 – 500 nm). The results are shown in

Figure S3 and the agreement is generally satisfactory (slope = 1.3 and 0.96, $R^2$ = 0.96 and 0.97, for 2015 and 2016 respectively).

The aerosol size distributions measured at the PEARL RidgeLab were compared against those measured at Alert, Nunavut located 480 km to the northeast (Figure S2), where the surface measurements were conducted at the Dr. Neil Trivett

Global Atmosphere Watch Observatory (82.5° N, 62.3° W), 210 m above sea level. At this site, particle size distributions between 10 and 487 nm were measured using a SMPS (TSI 3034) (Leaitch et al. 2013). Details of the aerosol sampling inlet at Alert are described in the previous work of Leaitch et al. (2013) and Leaitch et al. (2018).

## 2.2 Aerosol Mass Spectrometer

Between 26 July and 8 September 2015, a quadrupole aerosol mass spectrometer (AMS, Aerodyne Research Inc.)

measured the chemical composition of submicron non-refractory aerosol particles at the PEARL RidgeLab (Canagaratna et al. 2007). Both hourly bulk and size-resolved concentrations were measured by switching between mass spectrometry (MS) mode and particle time-of-flight (PToF) mode, which provides quantitative measurements in the range of 50 to 1000 nm (aerodynamic diameters). All data were analyzed using standard AMS software (AMS Analysis Toolkit v1.43) with Igor Pro v6.3.7.2 (WaveMetrics). The instrument was calibrated multiple times during the measurement period with 300 nm diameter

ammonium nitrate particles to determine the ionization efficiency. The aerodynamic diameter was calibrated using polystyrene latex spheres at 80, 125, 240 and 300 nm. There are two important limitations to the size resolved AMS measurements reported here. Firstly, it should be noted that the extrapolation of the aerodynamic diameter calibration below 80 nm is not well constrained, so particle size data below this diameter should be considered qualitative rather than quantitative. Secondly, the AMS inlet has less than 100% transmission efficiency below aerodynamic diameters of 70 nm,

although there is still substantial transmission of particles down to diameters of 30 nm. Filtered air was sampled every day to establish the air beam corrections. Aerosol mass measured by the AMS was corrected for the instrumental collection efficiency using the method of Middlebrook et al. (2012). The collection efficiency (CE) varied between 0.45 and 0.86 with the increases in CE corresponding to periods when aerosol sulfur was present in its acidic forms (sulphuric acid and ammonium bisulphate) rather than as ammonium sulphate. Vacuum aerodynamic diameters measured by the AMS were

converted to physical diameter under the assumption that the particles were spherical, contained no voids, and had a density of 1.25 g cm$^{-3}$ (DeCarlo et al. 2004). This density is typical for ambient organic aerosol (Middlebrook et al. 2012) and was selected for this study since the analysis was focused on the particle composition during the predominantly organic aerosol growth events.

To evaluate the accuracy of the AMS measurements, they were compared to the mass concentration of particles

having a diameter of less than 1 µm (PM$_1$) from the combined SMPS and OPC measurements. Applying the density calculated from the AMS data to the particle size distribution, a linear regression analysis of the AMS PM$_1$ mass concentration versus that calculated from the combined SMPS and OPC measurements resulted in a correlation coefficient of

0.89 and a slope of 1.16. These values confirm that the collection efficiency algorithm from Middlebrook et al. (2012) was reasonable.

## 2.3 Meteorological Data

Radiosondes (Vaisala RS92-SGP) launched from sea level at the Eureka Weather Station provided different meteorological parameters for altitudes both below and above the PEARL RidgeLab. The radiosondes are launched every 12 h by ECCC meteorological technicians, and the reported data were obtained from the University of Wyoming, Department of Atmospheric Sciences' Upper Air Data Website (http://weather.uwyo.edu/upperair/sounding.html). While the resulting measurements provide a means to evaluate the vertical temperature profile, and thus whether the PEARL RidgeLab at 610 m was located within or above the inversion layer, caution must be taken in interpreting the results due to a number of considerations: (1) the ECCC Weather Station is located approximately 11 km from the PEARL RidgeLab, (2) the complex terrain in the region, and (3) the radiosondes do not necessarily fly straight up and can meander significantly after launch because of the wind direction. Therefore, the radiosonde measurements do not necessarily reflect the vertical temperature profile near the PEARL RidgeLab.

## 2.4 Back-Trajectory Analysis

Air mass histories were computed using the FLEXible PARTicle (FLEXPART (Stohl et al. 2005)) Lagrangian-dispersion model. The tracer particles are inert and non-interacting and are released from the position of the PEARL RidgeLab at an altitude between 610 m above sea level. Backward dispersion runs were initialized by releasing an ensemble of 6000 air-tracer particles over a 6 hour period around the times corresponding to the beginning of the growth events listed in Table 1. The same parameters were used for Alert, except the fact that Alert is at 210 m and not at 610 m like the PEARL RidgeLab. FLEXPART was run in backward mode for 6 days driven by meteorological data from the National Centers for Environmental Prediction (NCEP) Climate Forecast System (CFS V2) 6 hourly product (Saha et al. 2014) to calculate the spatially resolved potential emissions sensitivity, which is proportional to the residence time of a tracer above a given grid cell. Potential emissions sensitivity represents the amount of time that an air mass is influenced by emissions within a given grid cell during the duration of the FLEXPART run. In this study, the potential emissions sensitivity is time-integrated over a period of 6 days before the particle release time.

## 3. Results and Discussion

## 3.1 Summertime Aerosol Size Distributions

## 3.1.1 Observations at Eureka and Alert

Figure 1 shows the aerosol size distributions measured at the PEARL RidgeLab and at Alert for 16 June to 26 September 2016. Particle growth events were evident at both sites. In total, 34 events with elevated concentrations of small

particles (< 20 nm diameter) were observed at the PEARL RidgeLab during this period, 22 of which were followed by growth lasting between 2 to 6 days. It is important to note that the local anthropogenic emissions should be completely negligible due to the extremely remote position of the site. The electricity for the PEARL RidgeLab is generated by a small power plant located 11 km from the site and there is no indication from the measurements that the site is significantly

influenced by emissions from the power plant or the Eureka Weather Station. The sudden appearance of Aitken mode particles is consistent with previous field observations performed in the Canadian Arctic during research flights and cruises (Chang et al. 2011; Leaitch et al. 2013; Willis et al. 2016; Collins et al. 2017). While the sources of these particles remains poorly understood, this previous work suggested that the formation and growth of ultrafine particles may be due to marine biological activity and the oxidation of DMS and volatile organic compounds (VOCs). The sustained particle growth

observed at the PEARL RidgeLab and at Alert, as well as in the previously published work cited above suggests that there is a significant atmospheric reservoir of chemical compounds with volatilities that are low enough to partition to a condensed phase and could thus also be contributing to the nucleation process. Nevertheless, it is not possible to rule out primary marine emissions as a source of particles that provide the necessary surface area for condensing gases (Leck and Bigg 2005a). During certain events that exhibit the appearance of Aitken mode particles and subsequent growth, there are also

signs of successive events that merge into the growth events from previous days, consistent with other observations in the Arctic (Collins et al. 2017).

Despite being almost 500 km apart, the particle growth events occurred at similar times at both the PEARL RidgeLab and Alert (Figure 1). While simultaneous nucleation events at sites as far apart as 350 km have been observed in continental regions where $SO_2$ concentrations are high (Jeong et al. 2010; Crippa and Pryor 2013), to our knowledge this

work is the first time such a correlation of specific events has been observed in the Arctic, although monthly averages have been previously compared (Freud et al. 2017). It is also important to note substantial topographic barriers exist between the two stations that are located on opposite sides of the Arctic Cordillera, which hinders direct passage of air masses between the two sites (see discussion of back-trajectories in Section 3.1.3). The particle number concentrations measured at the two sites for diameters between 10 and 487 nm are similar (Figure 2a), and the number concentration of particles between 20 and

70 nm at the two sites shows a moderate correlation with a correlation coefficient of 0.61 (Figure 2b). These results confirm that the growth events have a tendency to occur at similar times at both sites, demonstrating that conditions can exist in the Arctic that are favourable for aerosol growth over distances of at least 500 km.

Similar to Figure 1, the aerosol size distributions at the PEARL RidgeLab and Alert were measured for a portion of summer 2015 (26 July to 26 September 2015) as shown in the bottom panels of Figure 3. Again there is a clear correlation of

30 Aitken mode particles and their subsequent growth at the two sites, leading to the conclusion that the similarities in the growth events at the two sites are not specific to 2016.

In order to evaluate the influence of the appearance of small particles and growth events on the particle number concentrations at the two sites, the total concentration and the concentration of particles with a size between 10 to 100 nm, measured by the SMPS are summarized in Figure 4 for 27 Jul – 9 Sep 2015 and 2016. The particle concentrations are similar

at both sites and for both periods. The one exception is that the 90[th] percentile was higher for Alert in 2015, which was driven by two events with especially elevated particle concentrations. Coinciding events were observed at Eureka, but the particles concentrations were much lower. The reason for the elevated concentrations at Alert but not at Eureka is unknown. It is important to note that for 2016, the median is approximately $50 - 100$ particles cm$^{-3}$ higher than the results shown in Figure 4 if data from 16 Jun – 26 Sep 2016 are analyzed instead. This can be explained by the fact that the total duration of growth events was longer in June and July compared to events occurring in August and September.

### 3.1.2 Case Studies of Aerosol Growth Events

To further analyze the growth events and periods with elevated concentrations of ultrafine particles, two different sets of case studies were selected comprising 5 (Table 1) and 28 events (Table S1). The latter represents all the growth events observed during the measurement period (22 events during 2016 and 6 events during the shorter 2015 period), and the smaller set of 5 was used to calculate growth rates. This subset of events was chosen because they were distinct, without overlap with preceding or subsequent growth events and exhibited relatively smooth growth curves. The remaining 23 growth events were sometimes interrupted, presumably due to changes in air mass origin, or consisted of several events overlapping each other. All of the 5 growth events presented in Table 1 represent complete and smooth growth events that were suitable for calculating growth rate. For the smaller set of case studies near Eureka, the measured particle size distributions are shown in Figure 5, along with the temperature profiles measured using radiosondes launched from the Eureka Weather Station (Figure 6). Initial aerosol growth rates were calculated following previously published methods (Kulmala et al. 2004; Hussein et al. 2005; Salma et al. 2011). Briefly, the SMPS size distributions were fitted with a multi-mode log-normal distribution, and then a linear regression analysis was performed on the geometric mean of the Aitken mode as a function of time for particle diameters between $10 - 30$ nm. The initial growth rates calculated for this study are given in Table 1. Looking more closely at the meteorology of the five growth events, one can evaluate the optimal conditions that favor the presence of the growth events at the PEARL RidgeLab. In particular, the absence of an inversion below the PEARL RidgeLab would correspond to air masses measured at the site that are less photochemically aged and more influenced by local and possibly marine sources. In contrast, if the sources of aerosol mass during growth are the sea or the land surface and within the stable stratification, then these influences are expected to be less important when an inversion is present below the PEARL RidgeLab. Figure 6 demonstrates that while temperature inversions that terminate with a maximum below 600 m (the altitude of the PEARL RidgeLab) do sometimes exist before or at the beginning of a growth event, the presence of such inversions is infrequent and often weak (less than 2°C). These inversion conditions will thus result in air masses measured by the instruments at PEARL that are directly influenced by local emissions near or below the PEARL RidgeLab.

To more systematically analyze all the growth events for the summers of 2015 and 2016 at the PEARL RidgeLab, a histogram of the number of events binned by the average inversion temperature (i.e. the temperature at the top of the inversion minus that at the bottom) during each event is plotted in Figure 7. (All the growth events used in creating Figure 7

are summarized in Table S1 and the particle size distributions are shown in Figure S4.) There was a clear tendency for particle growth events (and presumably nucleation) to occur when the inversion was weak or absent. Furthermore, we conducted a similar analysis for six periods when particle concentrations were low and for six periods with a persistent accumulation mode (summarized in Table S2 and Figures S5 and S6). Figure 7 shows that the average inversion temperature during the growth events ($0.3 \pm 0.7$°C) was very similar to that during the selected periods with low particle concentration ($0.3 \pm 0.2$°C), whereas the average inversion temperature during periods with a persistent accumulation mode and elevated particle concentrations was much higher ($2.5 \pm 1.2$°C). The results shown in Figure 7 imply that growth events occur at the PEARL RidgeLab when the inversion is weak because, firstly, those periods correlated with low particle surface area concentrations and corresponding condensation sink in the marine boundary layer air which allows particle nucleation to occur, and secondly, the site was possibly influenced by more recent surface emissions that were less photochemically aged compared to air aloft. In contrast, when the inversion was strong, the aerosol and aerosol precursor species were more chemically aged due to slower transport into the free troposphere and thus the existing particles had already grown to sizes corresponding to the accumulation mode. A few growth events were observed when the temperature inversion was larger, which may be due to the fact that the radiosondes were launched at the Eureka Weather Station located 11 km to the southwest of the PEARL RidgeLab. Thus, the temperature profile measured by a radiosonde may not be fully representative of that at the RidgeLab. Generally speaking, the observations reported here are consistent with previous work (Willis et al. 2016; Collins et al. 2017) suggesting that similar events measured in the Canadian Arctic are attributable to marine sources.

Previous studies have characterized aerosol growth rates in remote regions including the Arctic. In particular, Collins et al. (2017) reported growth rates ranging from $0.2 - 15.3$ nm h$^{-1}$ during two research cruises conducted in the Canadian Arctic and calculated a corresponding average growth rate of $4.3 \pm 4.1$ nm h$^{-1}$. Similarly, Nieminen et al. (2018) reported for Alert and Mt Zeppelin, Norway, that the average growth rates, between June and August, were 1.1 and 1.2 nm h$^{-1}$, respectively for the years $2012 - 2014$ and $2005 - 2013$. Moreover, Kolesar et al. (2017) observed an average growth rate of $1.8 \pm 1.5$ nm h$^{-1}$ for spring-summer marine air masses at Barrow, AK. In our study, growth rates ranged from $0.1 - 1.0$ nm h$^{-1}$ for the aerosols at the PEARL RidgeLab and at Alert, with an average rate of $0.5 \pm 0.3$ nm hr$^{-1}$ (Table 1). These values overlap with those reported in Collins et al. and are even more similar to those in Nieminen et al. and Kolesar et al. It should be noted that the size range used for calculating growth rates in our work ($10 - 30$ nm) is slightly different from that of Collins et al. ($4 - 20$ nm) and Nieminen et al. ($10 - 25$ nm). However, it is more likely that different condensable vapour concentrations or different environmental conditions (e.g. temperature, solar radiation, etc.) led to the variations in the observed aerosol growth rates. Lastly, the growth rates are similar for all 5 events analyzed in Table 1, which suggests that the atmospheric processes (e.g. the condensation of semi-volatile or low volatility vapors to the particle surfaces as discussed below) and conditions governing the growth events are similar for all the events in this study.

### 3.1.3 Back-Trajectory Analysis

To understand the influence of the air mass history on the occurrence of the growth events, back-trajectories were calculated using FLEXPART (Figure S7) for Eureka and Alert. (The particle size distributions for the analyzed events for Eureka and Alert are shown in Figures 5 and S8, respectively.) This calculation permits the precise evaluation of the spatial distribution of the potential emissions sensitivity at the beginning of each growth event. In general, these calculations show that the aerosols measured at the PEARL RidgeLab are mostly influenced by source regions located in the Canadian Arctic Archipelago, in Baffin Bay and to the north of Ellesmere Island. These results mostly coincide with the research reported by Collins et al. (2017), in which they observed high concentrations of ultrafine particles in these regions. Furthermore, the analyzed growth events generally have similar air mass histories for both the PEARL RidgeLab and Alert for a given event. The exception is GE 30 at the PEARL RidgeLab, which began on 25 June 2016, was more influenced by areas near and further north of Alert with a small contribution from the Nares Strait region. Interestingly, NASA Worldview MODIS images (https://worldview.earthdata.nasa.gov/) show that on 25 June 2016 (Figure S9) and for several preceding days, while the ocean in these regions was mostly covered in sea-ice, the potential emissions sensitivity was still influenced by large areas of open water. In conclusion, growth events can occur within air masses with different back-trajectories, as also reported by Collins et al. (2017), although the potential emissions sensitivities for the five growth events shown in Figure S7 have a substantial amount of overlap. Furthermore, we conducted a similar back-trajectory analysis for the six periods when particle concentrations were low and for the six periods with a persistent accumulation mode as described above in Section 3.1.2 and summarized in Table S2. The results are shown in Figures S10 and S11. There are no clear differences between the back-trajectories for the different types of periods and the growth events with almost all back-trajectories showing substantial potential emissions sensitivities over continental and marine regions mostly within the Arctic.

### 3.2 Aerosol Bulk and Size Resolved Chemical Composition

AMS measurements of aerosol chemical composition and mass concentration for the summer of 2015 are shown in Figures 3a and 3b, where the $PM_1$ mass concentrations include the four dominant types of non-refractory aerosol. During two major growth events in July and August (GE 3 and GE 6), it can be seen that the aerosol organic fraction represented a large majority of the aerosol mass (Figure S12). In contrast, later in the summer (approximately 30 August 2015 to 5 September 2015) there is a period of larger particles when the mass concentration of sulphate is higher than the organic component. While iodine may contribute to particle nucleation, the low resolution of the quadrupole AMS prevents quantification of iodine. Given the low signal at m/z 127 in this study during growth events, we believe that iodine is at most a minor contributor of mass to Aitken mode particles.

While sulphate is a key contributor to nucleation at lower latitudes and is an oxidation product of dimethyl sulphide (DMS) from marine emissions, relatively little sulphate is observed during growth events. At the same time, the observation of relatively high bulk organic aerosol concentrations during Arctic summer is consistent with previous analyses of organic

aerosol functional groups in samples collected at Alert (Leaitch et al. 2018). Since the bodies of water in the vicinity of Eureka were relatively open and not covered in sea ice during this period, the water may have been a source of precursors to the observed aerosol mass, which would be consistent with the correlation of the occurrence of growth events and the breakdown of the temperature inversion below the altitude of the PEARL RidgeLab. It is possible that methanesulphonic acid (MSA), an atmospheric oxidation product of DMS, could be contributing to the overall organic mass (Park et al. 2017). However, the mass spectra for GE 3 and GE 6 are very different from the MSA spectrum measured in the laboratory (Figure S13) (Phinney et al. 2006). In particular, the relative intensity of m/z 79 is much lower in the ambient spectra. Furthermore, in the MSA spectrum the sulphate fragments at m/z 48 and 64 are much greater than the organic fragments at m/z 43 and 44 whereas these organic fragments had greater intensity in the ambient spectra. Lastly, the correlation coefficient for the two average ambient mass spectra (R = 0.84) is much higher than the correlation coefficient between each ambient spectrum and the spectrum measured for MSA (R = 0.60 and 0.48 for GE 3 and GE 6 versus MSA, respectively). When taken together, these differences between the ambient and MSA mass spectra indicate that other organics were contributing to the aerosol mass besides MSA. However, the size distribution of m/z 79 during GE 6 shows some signal below 100 nm, suggesting that MSA could be present in Aitken mode particles during at least some growth events (Figure S14). To further investigate these findings, the AMS fragmentation table was also modified to separately quantify MSA following Phinney et al. (2006), but the concentration of MSA was generally at or below the detection limit (0.021 μg/m$^3$ as determined by multiplying by 3 the standard deviation when the AMS was sampling through a filter). Based on this detection limit, the MSA concentration was 5% or less of the total organic and sulfate mass concentration during the two measured growth events, GE3 and GE6.

In addition to the measurements of the bulk aerosol composition shown in Figure 3, the dependence of the composition on the particle size is shown in Figure 8. The PToF data revealed that the smallest particles sampled by the AMS (between 50 and 80 nm in diameter) during the two different growth events were enriched in organics, with little to no sulphate. Between 80 and 1000 nm, the measured aerosol composition changes depending on the particle size and the larger aerosol particles contain a greater fraction of sulphate. The higher concentration of sulphate in the larger particles is most likely explained by the presence of a distinct accumulation mode having a history and source different from the Aitken mode aerosols.

To further evaluate the organic aerosol composition only, two important fragments in the measured mass spectra of the organic aerosol, m/z 43 and 44, are plotted in Figure 8e and Figure 8f. For organic aerosol, m/z 44 corresponds to the concentration of carboxylic acids, whereas m/z 43 correlates with other oxygen containing functional groups in the particle phase (e.g. alcohols). The size-resolved measurements of these fragments show that the organic composition varied with particle size. The smallest particles were less oxidized, with the fraction of carboxylic acids increasing with the particle diameter between 80 and 400 nm. While this relative trend was observed for both growth events, the absolute ratios of m/z 43 to m/z 44 were different, which indicates some variation in the amount of organic aerosol oxidation during growth events.

To complement the analysis of the organic aerosol composition shown in Figure 8, the fractions of the total organic mass measured at m/z 43 and at m/z 44 (abbreviated as $f_{43}$ and $f_{44}$) are plotted against each other in Figure 9, in which the

data are colored as a function of time. For reference, the size-resolved $f_{43}$ and $f_{44}$ are also shown in Figure S15. Lambe et al. (2011) demonstrated in a series of laboratory studies that secondary organic aerosol (SOA) formed from a variety of different precursors, both anthropogenic and biogenic, falls within a well-defined space in the $f_{44}$ versus $f_{43}$ plot, shown as the black triangle in Figure 9. The organic aerosols measured at the PEARL RidgeLab have $f_{44}$ and $f_{43}$ ratios that are consistent with the previous work of Lambe et al. and others (Ng et al. 2011). Furthermore, the mean $f_{44}$ and $f_{43}$ for the seven hours at the beginning and end of GE3 and GE6 are included in Figure 9 in order to evaluate the overall change in SOA composition.

From Figures 8e and 8f, we speculate that the smaller and larger particles are reflective of SOA formed earlier and later during the two growth events, respectively. The greater fraction of the signal at m/z 44 in the accumulation mode relative to the Aitken mode would thus represent increased oxidation and greater production of carboxylic acids as the events progressed. This would be consistent with the slight to moderate increase in $f_{44}$ observed in Figure 9 throughout both events. However, there was insufficient signal in our measurement to directly observe a change in $f_{44}$ in the Aitken mode aerosols to prove that oxidation actually increased. It is entirely possible that these observed differences were due to larger-scale processes that changed the overall aerosol population without SOA formation. Moreover, we emphasize that our results are for a very limited data set and further analysis of SOA composition during additional growth events using $f_{44}$ and $f_{43}$ would be necessary to confirm our observations and speculations.

In summary, the concentration of oxygenated organics as well as the presence of small organic particles measured by the AMS together suggests that SOA formation contributes to particle growth measured at the PEARL RidgeLab. While the origin of the SOA precursors is unknown, it can be concluded that the organic composition is inconsistent with SOA formation from MSA alone. Recently published work has suggested that marine microbial processes may be an important source of these VOCs (Collins et al. 2017). Abiotic heterogeneous processes in the marine boundary layer may also be a source of oxidized VOCs as indicated in several laboratory studies (Bruggemann et al. 2017; Chiu et al. 2017). Consistent with this previous research, the first growth event observed at the PEARL RidgeLab in 2016, coincided approximately with the melting of the sea ice in the Slidre Fjord and the Eureka Sound located to the south and west of the PEARL RidgeLab. Observations of the sea ice taken from the PEARL RidgeLab are shown in Figure S16. The pictures show, for summer 2016, the sea ice during the first (25 June 2016) and last (10 September 2016) growth events. Also shown are two additional images. One is of the first time that open water was observed (14 July 2016), and the other is the last available image of Eureka Sound and Slidre Fjord for 2016 (28 September 2016) before polar sunset made it too dark for photographs. On 25 June, it is not possible to see regions of open water during the first growth event. However, open water was observed 300 km to the south of Eureka on the same day in NASA Worldview MODIS images (Figure S9). Moreover, the NASA Worldview MODIS images show open water on 7 July in Eureka Sound and Slidre Fjord. It should also be noted that the first growth event occurred much later than polar sunrise on 21 February 2016. During the last observed growth event (10 September 2016), there was very little or no sea ice, and open water was persistently observed until the end of September. This is consistent with measurements of DMS and MSA at Alert which show relatively high concentrations persisting into

September (Sharma et al. 2012; Leaitch et al. 2013). Given these observations, there is a possible relationship between the onset of the growth events and the melting of the sea ice in the region around Eureka. This would be consistent with back-trajectory analyses showing that newly formed particles measured during summertime cruises in the Arctic Ocean are associated with air that has experienced more open water or melting sea ice regions (Heintzenberg et al. 2015; Dall'Osto et al. 2017). In contrast, the decrease in the number of events in September was more likely due to the lack of solar radiation, as shown in Figure S17 and previous measurements at Alert (Sharma et al. 2012), which limits photochemistry. This finding supports the recent suggestion that photochemical processing of emissions from the ocean may be a source of ultrafine particles in the Arctic (Collins et al. 2017).

## 4. Conclusions

In this study, particle growth events were characterized during the summers of 2015 and 2016 at the PEARL RidgeLab (in Eureka, Nunavut, Canada) as well as at Alert, Canada. Both sites are located on Ellesmere Island separated by a distance of 480 km providing an opportunity to evaluate the growth events on a regional scale for the complete 2016 summer season as well as for a portion of the 2015 summer season. During both years, frequent growth events occurred and these events were correlated between the sites. In addition to the concomitant events, the particle concentrations measured at Alert and the PEARL RidgeLab were similar, with the $10^{th}$, $25^{th}$, $50^{th}$, $75^{th}$ and $90^{th}$ percentiles not varying by more than a factor of 1.67 suggesting the growth events were not isolated local events. Additionally, the mean particle growth rate from a subset of events at the PEARL RidgeLab and at Alert was $0.5 \pm 0.3$ nm hr$^{-1}$. In total, the measurements of the particle number size distribution support the conclusion that particle nucleation and growth events can occur over spatial scales of at least 500 km in the Canadian Arctic Archipelago. Previous work in the summer time Arctic found that particles smaller than 50 nm could be contributing to cloud droplet activation (Leaitch et al. 2016). The growth of small particles to diameters larger than 60 nm observed in our study could therefore make them an important contributor to CCN, ultimately impacting the radiation balance and hydrologic cycle.

Moreover, in this study AMS measurements showed that particles between 50 and 80 nm in diameter during two observed growth events were predominately organic. The amount of oxidation of the organic fraction also changed with particle size, with the ratio of m/z 43 to m/z 44 increasing for smaller particles sizes, which is consistent with a greater fraction of non-acid oxygenates relative to carboxylic acids. Overall, our limited AMS measurements support the conclusion that condensation of organic vapors contributed to particle growth.

It has been recently suggested that secondary organic aerosols formed from VOCs emitted by marine sources may be an important source of ultrafine particles in the Arctic during summertime. The results of the research presented here are consistent with this possibility. In addition to the SMPS and AMS measurements discussed above, the growth events were most likely observed at the PEARL RidgeLab when the inversion was non-existent or weak, allowing the site's instruments to sample the boundary layer which had a low aerosol condensation sink as well as presumably fresh marine emissions. Finally, the onset of the growth events in 2016 coincided more with the opening of the sea ice near the PEARL RidgeLab,

rather than polar sunrise. However, future work should focus on the incorporation of more sea ice data as well as further gas phase measurements to understand the timing and chemical processes driving particle nucleation and growth in the Arctic.

## Author Contributions

S.T., P.L.H and R.Y.W.C. performed the field measurements at the PEARL RidgeLab and authored the manuscript. J.C.P. analyzed the radiosonde data to obtain the temperature profiles. J.O.B. carried out the particle loss calculations. E.L. and K.S. performed the back-trajectory calculations. W.R.L., S.S. and F.K. provided aerosol size distribution data for Alert. C.J.C provided radiation data for Eureka, P.F. assisted with the measurements at the PEARL RidgeLab.

## Acknowledgements

This work was partially supported by the Université de Montréal and the Natural Science and Engineering Research Council of Canada (Discovery Grant RGPIN-05002-2014, Discovery Grant RGPIN-05173-2014 and Climate Change and Atmospheric Research (CCAR) Program funding awarded to the Probing the Atmosphere of the High Arctic (PAHA) project led by PI James R. Drummond). Support from Environment and Climate Change Canada (ECCC) for the Arctic field work performed at both Eureka and Alert is also acknowledged. The authors also acknowledge the Canadian Network for the Detection of Atmospheric Change (CANDAC) staff for maintaining the infrastructure at the PEARL site, Canadian Forces Station Alert for the maintenance of Alert base, and the use of the FLEXPART Lagrangian dispersion model (https://www.flexpart.eu/wiki/FpDownloads) and the Pflexible Python module (https://bitbucket.org/jfburkhart/pflexible) developed by John F. Burkhart, which were modified here to plot the FLEXPART sensitivities in this paper. The NCEP CFS data used are listed in the references. Radiation measurements have received support from ECCC, CANDAC and the Arctic Research Program of the NOAA Climate Program Office. Finally, the authors are indebted to James Sloan, Asan Bacak, Thomas Kuhn and Richard Damoah for their early efforts in installing and maintaining the AMS at PEARL and to James Drummond for his continued dedication in support of PEARL.

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

**Table 1.** Particle growth rates for five growth events during the summers of 2015 and 2016.

| Growth Event Number | Time period (UTC) | | | | Growth rate (nm/h) |
|---|---|---|---|---|---|
| | Start | | End | | |
| GE 3 (Eureka) | 2015-07-29 | 05:00 | 2015-07-30 | 11:00 | 0.420 ± 0.004 |
| GE 3 (Alert) | 2015-07-29 | 03:00 | 2015-07-30 | 12:00 | 0.50 ± 0.02 |
| GE 6 (Eureka) | 2015-08-02 | 04:00 | 2015-08-03 | 05:00 | 0.12 ± 0.08 |
| GE 30 (Eureka) | 2016-06-25 | 20:00 | 2016-06-27 | 14:00 | 1.01 ± 0.08 |
| GE 32 (Eureka) | 2016-07-04 | 05:00 | 2016-07-08 | 09:00 | 0.44 ± 0.01 |
| GE 38 (Eureka) | 2016-07-21 | 19:00 | 2016-07-25 | 17:00 | 0.352 ± 0.004 |

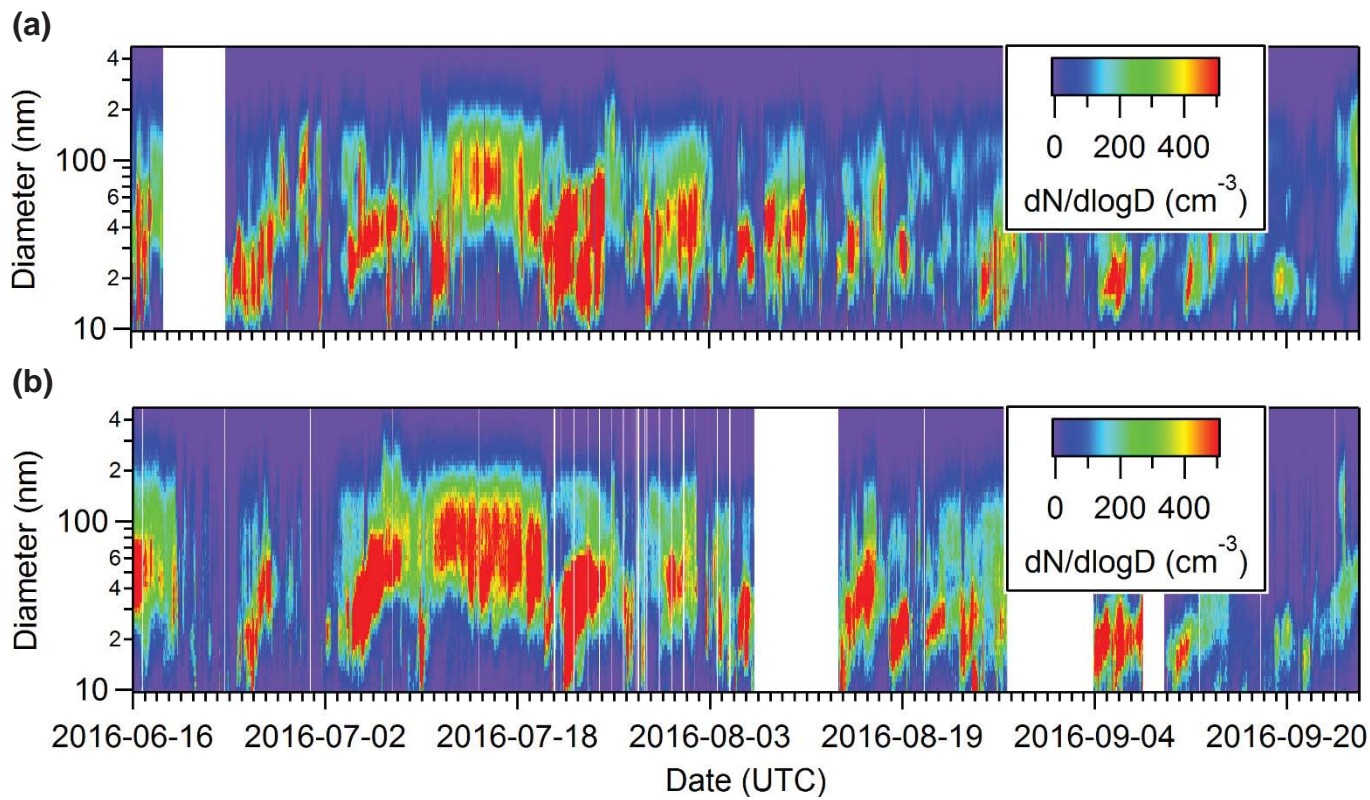

**Figure 1.** The size resolved particle concentration measured by SMPS instruments during summer 2016 in Alert **(a)** and at the PEARL RidgeLab **(b)** in the Canadian Arctic Archipelago. The sizes are mobility diameters measured by an SMPS, which are equal to the physical diameters under the assumption that the particles were spherical and contained no voids.

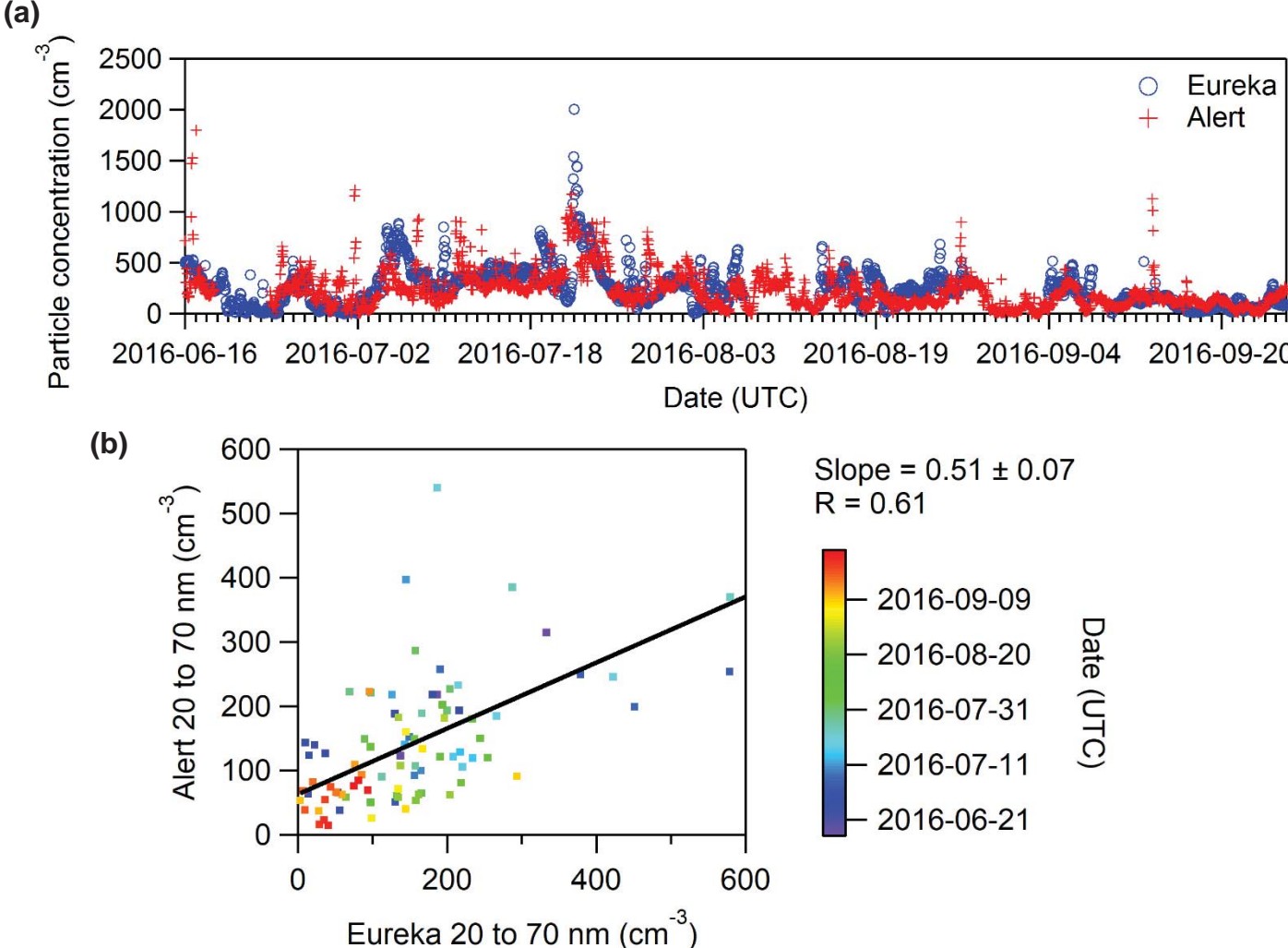

**Figure 2.** Total hourly particle number concentrations measured at the PEARL RidgeLab near Eureka and in Alert during summer 2016 for sizes between 10 – 487 nm **(a)**. Scatter plot showing the correlation of the particle number concentrations measured in Alert and near Eureka **(b)**. Note that the data in the scatter plot correspond to daily averages of particles with diameters between 20 and 70 nm.

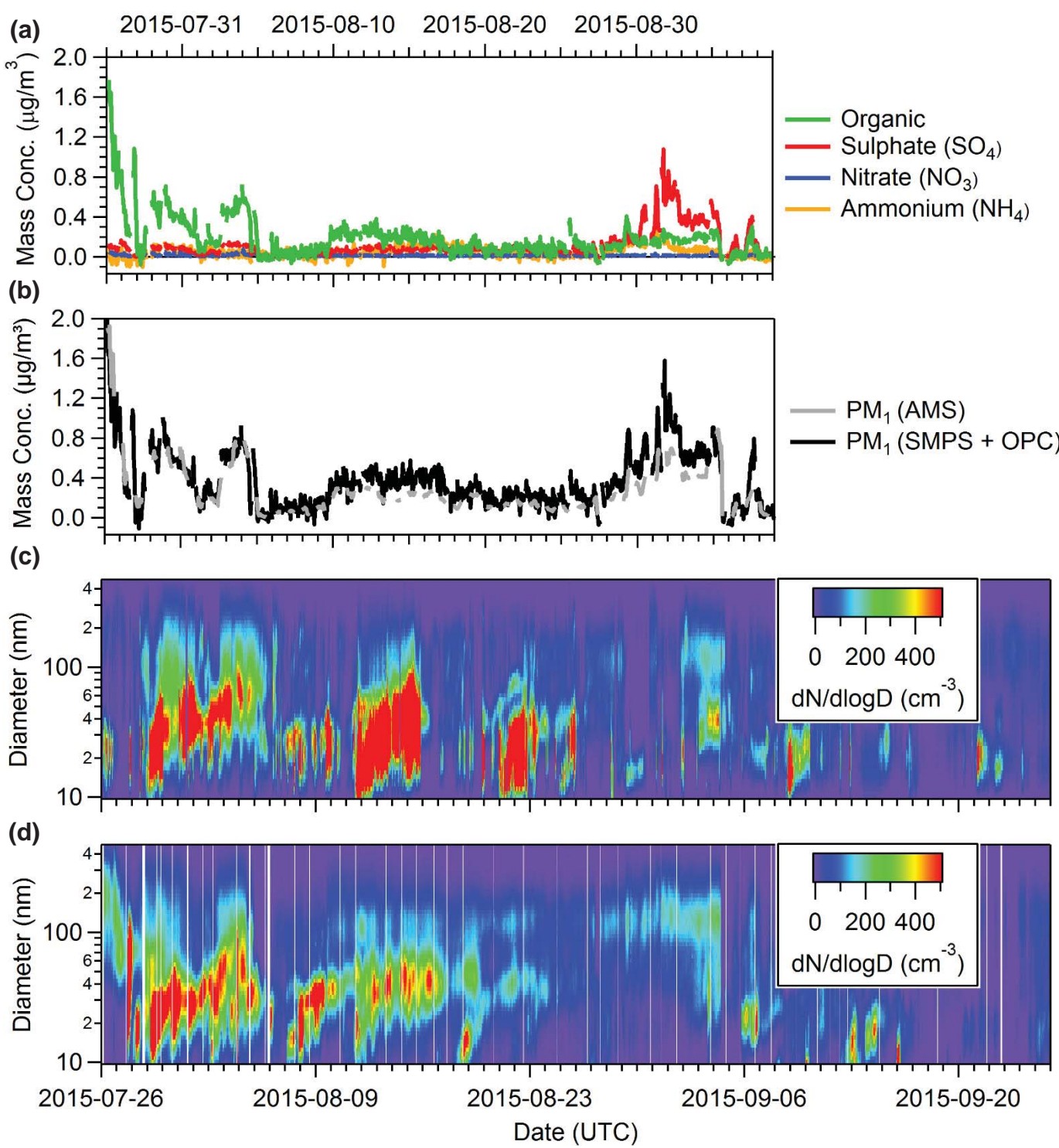

**Figure 3.** Aerosol mass spectrometry measurements of aerosol composition taken at the PEARL RidgeLab near Eureka **(a)**. The total concentration of non-refractory PM$_1$ aerosol measured by the mass spectrometer is also compared against the total PM$_1$ concentration measured by the SMPS and OPC, and exhibits good agreement with a linear regression analysis yielding a slope of 1.16 and a correlation coefficient of 0.89 **(b)**. In addition, the size resolved particle concentration measured by SMPS instruments during summer 2015 in Alert **(c)** and at the PEARL RidgeLab **(d)** are shown in the bottom two panels. The sizes are mobility diameters measured by an SMPS, which are equal to the physical diameters under the assumption that the particles were spherical and contained no voids. All data are plotted on the same time scale.

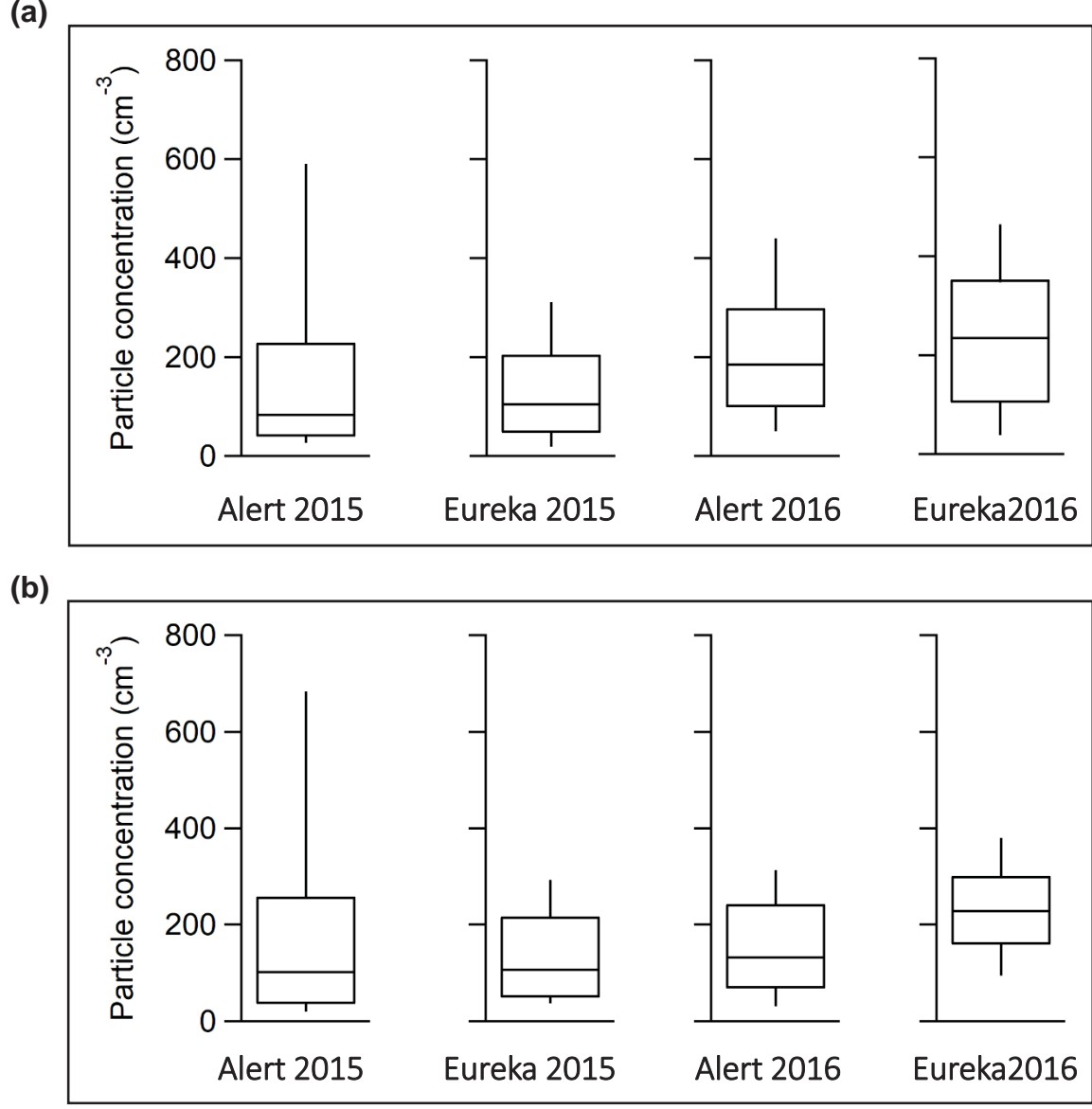

**Figure 4.** Box and whiskers plots of hourly particle number concentrations for diameters between **(a)** 10 – 487 nm, and **(b)** 10 – 100 nm measured during 27 Jul to 9 Sep 2015 and 2016 in Alert and at the PEARL RidgeLab near Eureka. The plots indicate the 10th, 25th, 50th, 75th and 90th percentiles.

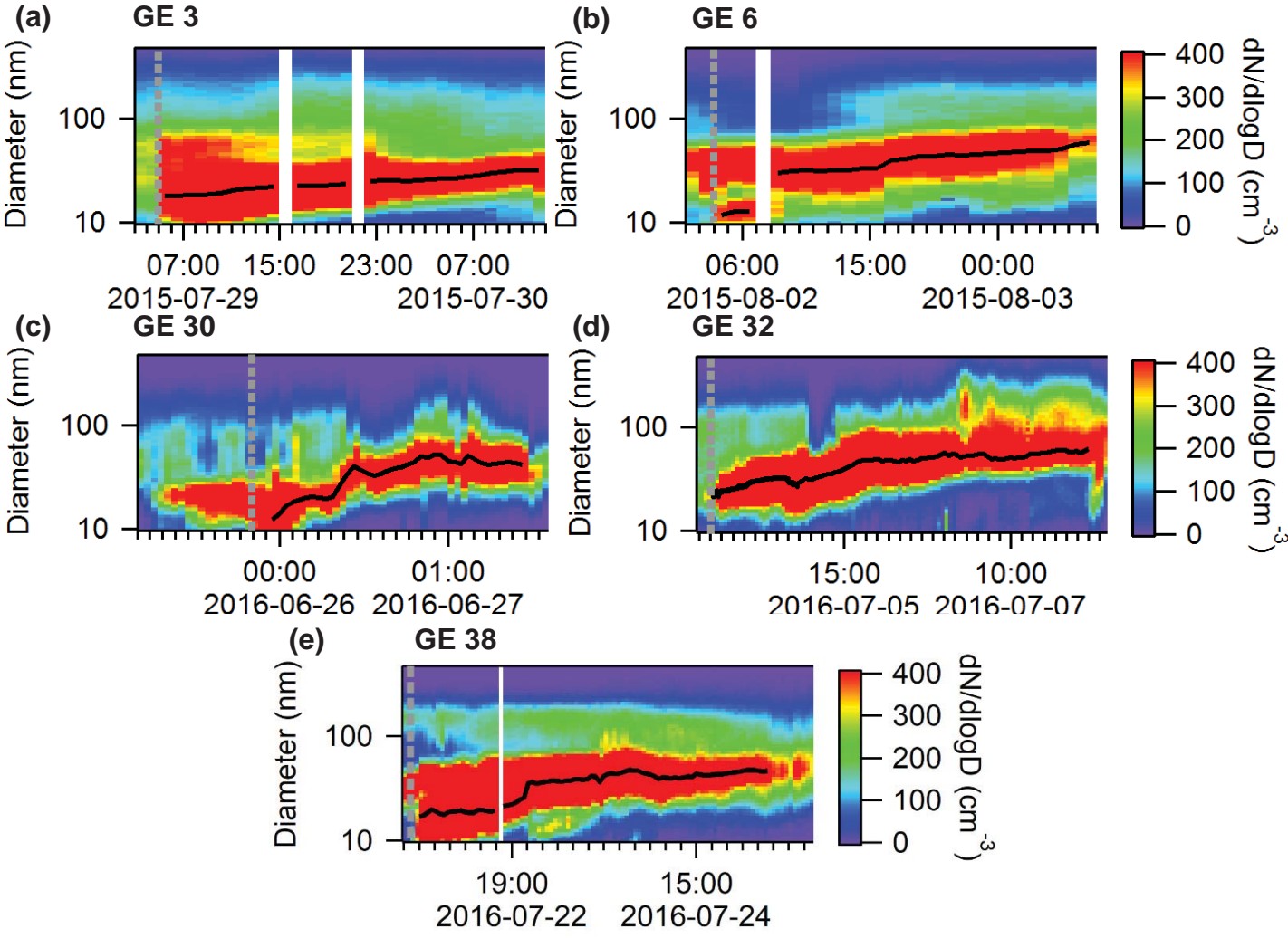

**Figure 5.** Five selected growth events near Eureka during the summers of 2015 and 2016. The grey dashed line indicates the start of each growth event and the black line indicates the Aitken mode diameter. The sizes are mobility diameters measured by an SMPS, which are equal to the physical diameters under the assumption that the particles were spherical and contained no voids.

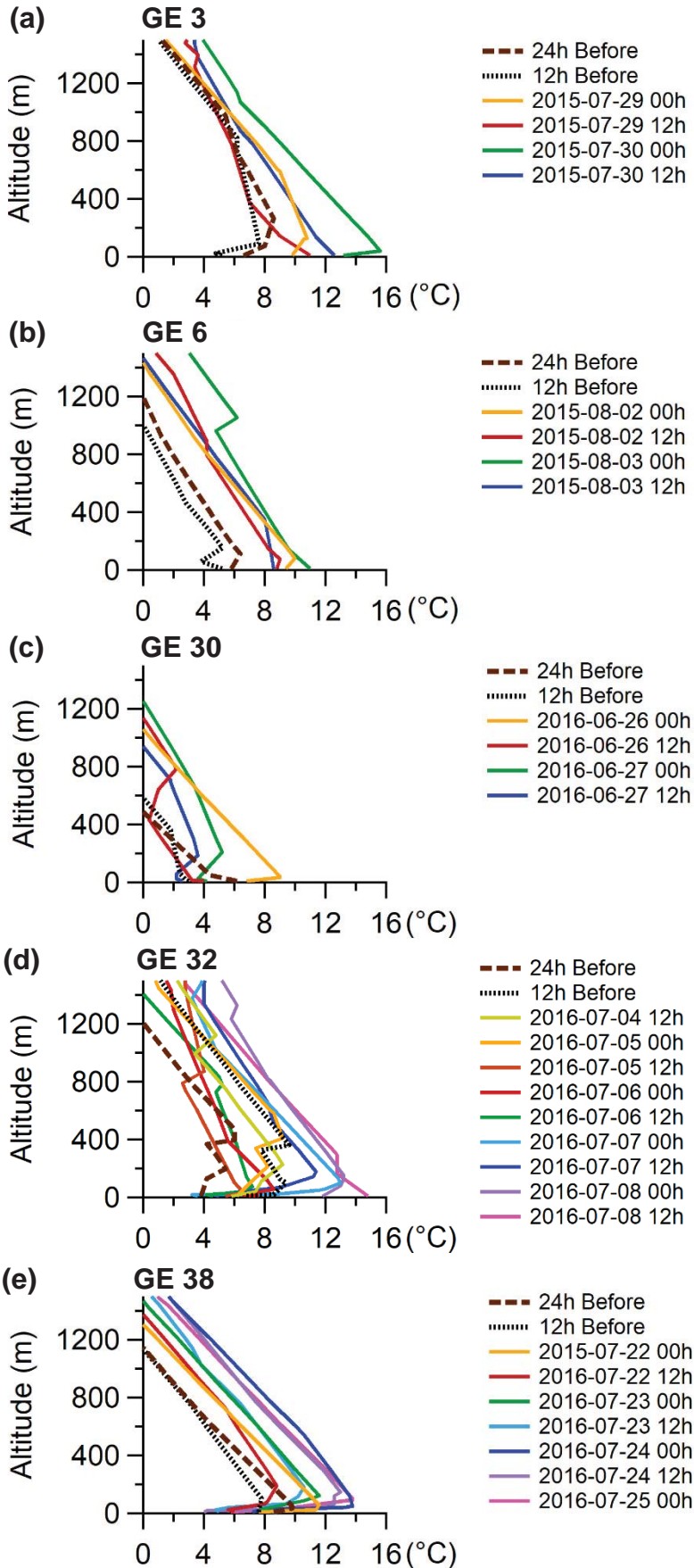

**Figure 6.** Vertical temperature profiles every 12 h during the 5 selected growth events that are shown in the previous figure. Data includes 24 h and 12 h prior to the start of the growth event.

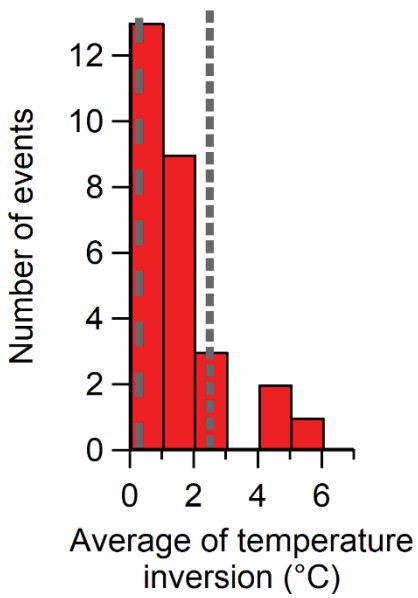

**Figure 7.** Histogram of the number of growth events near Eureka, binned by the average change in the temperature from 10 to 600 m above sea level. The average temperature change of each event is provided in Table S1, and was calculated from radiosonde measurements during 2015 and 2016, as shown in Figure 6. The dashed line indicates the average change in temperature during periods of low particle concentrations as shown in Table S2 and Figure S5 (0.3 ± 0.2 °C), and the dotted line indicates the average change in temperature during periods with a persistent accumulation mode as shown in Table S2 and Figure S6 (2.5 ± 1.2 °C). The values in parenthesis are the averages and their standard deviations.

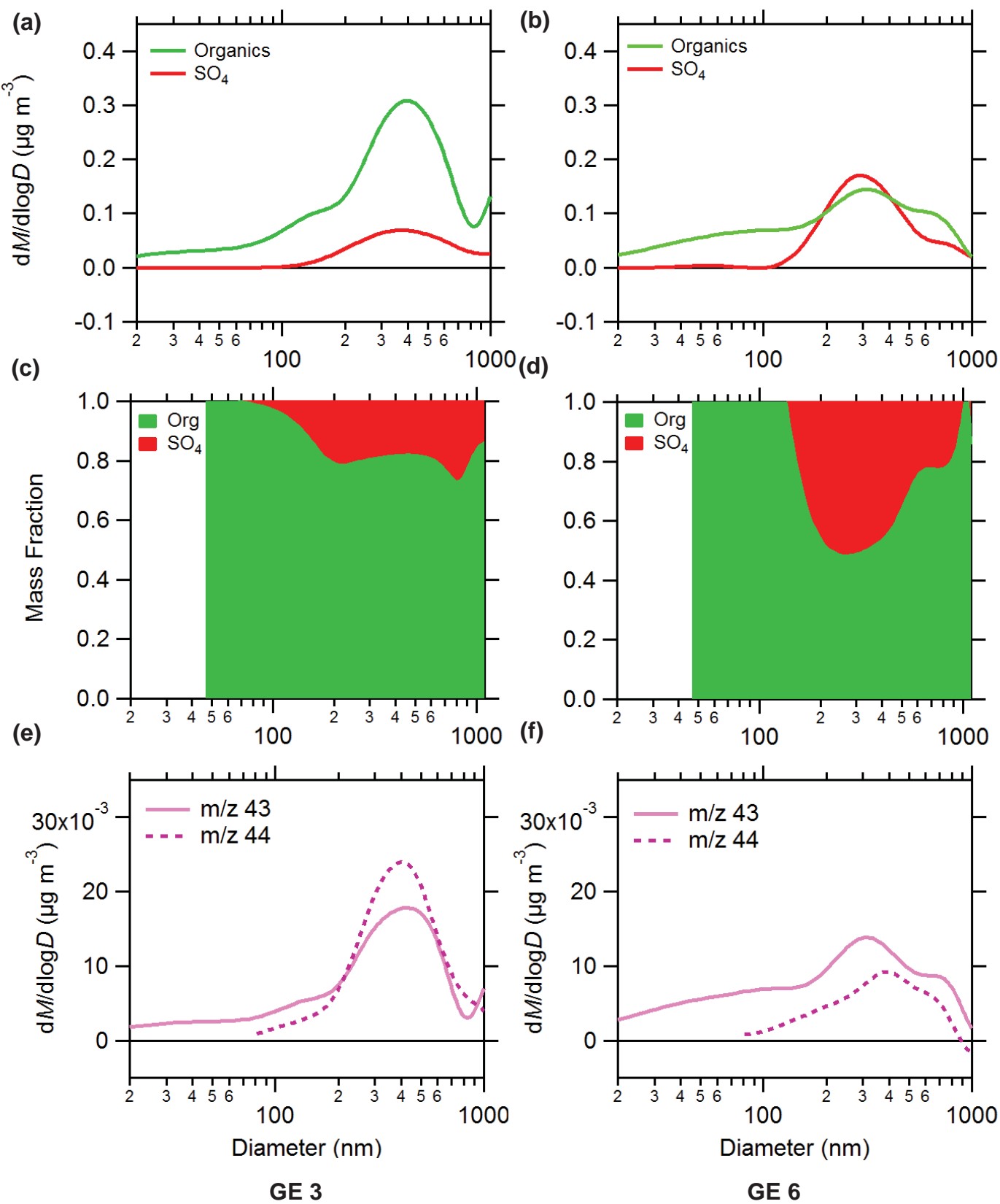

**Figure 8.** Size-resolved aerosol chemical composition for GE 3 **(a, c, e)** and GE 6 **(b, d, f)** measured near Eureka and averaged over the periods indicated in Table 1. The absolute organic and sulfate aerosol mass concentration **(a, b)**, the corresponding mass fractions **(c, d)**, and the nitrate-equivalent mass concentration of the m/z 43 and 44 fragments **(e, f)** are shown for each growth event. The data for m/z 44 is not shown below 80 nm due interference from gaseous $CO_2$. The uncertainties of a, b, e and f are 30 % and for c and d are 5 %.

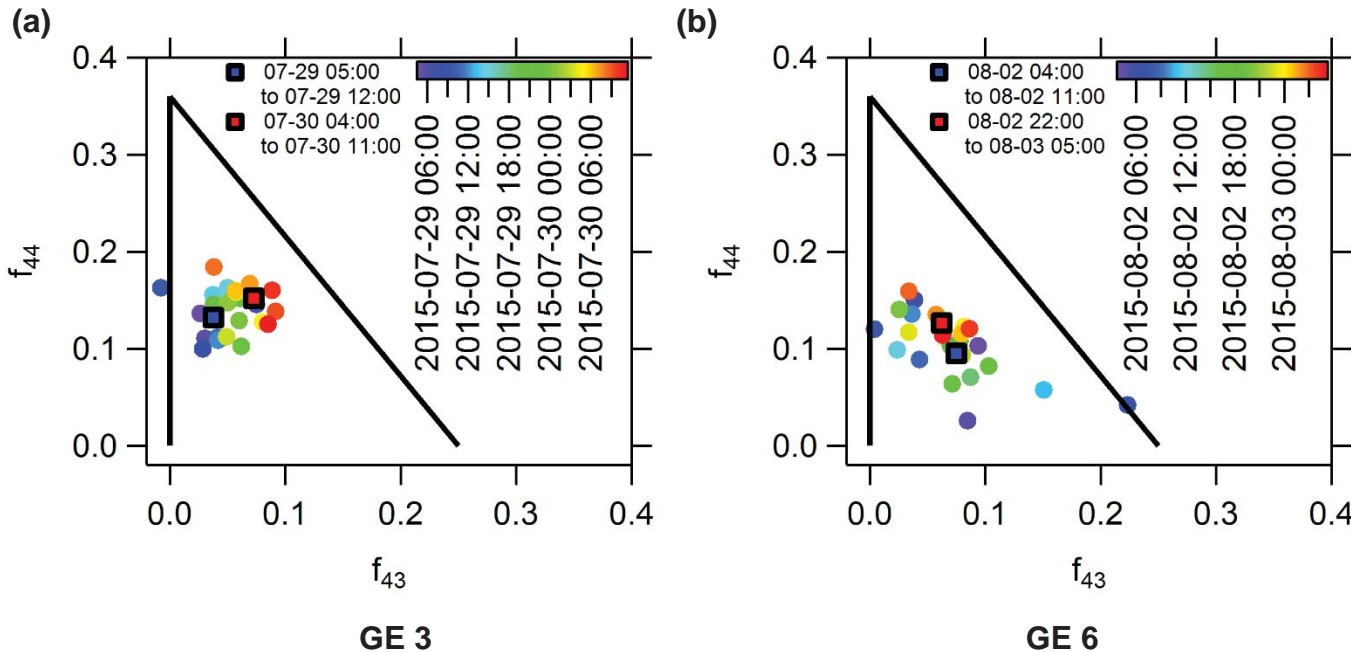

**Figure 9.** Scatter plot of the organic aerosol fraction measured at m/z 43 versus that measured at m/z 44 during GE 3 **(a)** and GE 6 **(b)** near Eureka. Also shown are the average values corresponding to the first seven and final seven hours of the growth event.