# Peer review of "Characterization of aerosol growth events over Ellesmere Island during summers of 2015 and 2016"

_Atmospheric Chemistry and Physics, 2018_

## Referee Comment (RC1) · Anonymous Referee #1 · 4 Jun 2018

Review of "Characterization of aerosol growth events over Ellesmere Island during summers of 2015 and 2016" by Tremblay et al.

This paper presents a detailed investigation into nucleation events in the Canadian arctic from summers 2015 and 2016, using aerosol size distributions and composition measurements. The paper is well written and the analysis is very thorough. So thorough in fact that it's in danger of being a little boring at times, but the conclusions are quite interesting. Overall the analysis and data are good and I recommend publication subject to the following minor corrections.

[Figure]

**Main comments**

One thing to consider for good measure is iodine-based nucleation which has been observed in Arctic marine summers before. You should mention it in your introduction and then probably rule it out for your measurements.

P3 paragraph 2- why such a focus on GEOS-Chem? What about other models? Or just models in general?

Section 2.1 a map would be really useful here, showing where all the field sites are and preferably the orography as well

P4 Paragraph 2- could you move all the discussion of the tubing etc to the supplement? It's good to be thorough but this doesn't contribute to the science of the paper. I'm sure you'll agree it's also not the most interesting paragraph ever written...

P4L23 Any idea why the difference in the OPC performance? How does that affect your AMS comparison?

P7L25 This is written strangely, it sounds like you are saying you got the numbers from table 1 from the 3 papers you reference, which isn't the case. I think it would be better something like *"Aerosol growth rates were calculated [say here very briefly how they are calculated and give the relevant reference(s)]. The initial growth rates from this study are included in Table 1."* It would be useful to comment on the meaning of this average growth rate as well, since it takes a lot more mass to go from 100nm to 101 than to go from 10 to 11. Is the number skewed towards being more representative of growth at any particular part of the size distribution?

P8L10 and Fig 6- can you add on the inversion strength for non-nucleation events and see if there is an obvious difference? That would strengthen your hypothesis that it's the lack of inversion that's creating conditions conducive to nucleation at the measurement altitude

P10L19 I'm not sure I agree there's a clear trend- there's a slight trend if you take

some pretty big averages but mostly there's just random noise. The average data points are not very far apart either. How does that compare to what you expect in this environment? I think adding some literature values to Fig 8 would be useful to add some context

P11L6 could you put the NASA worldview images in the supplement?

Finally, I think it would be useful to discuss the implications for of your findings for CCN concentrations in the Arctic.

**Technical/stylistic corrections**

I found the tenses very confusing throughout. Take for example the abstract. You say *"Measurements. . . .were taken during. . .2015 and 2016"*. You then switch to the present (incorrectly in my view) and say *"These events are observed beginning in June. . .marine sources are the primary cause."* It's not like you took data from 2 decades and can really talk in general terms about events that you expect to happen every summer. You have data from 2 years, so it's appropriate to refer to your measurements and things that you observed in the past. Then you talk about what the graphs show in the present. Other examples of incorrect tense are P2L16, P3L24, P4L18, but there are many more. Please be consistent and refer to things that happened in the past, in the past tense.

P7L23 preceding not proceeding

Figure 5 parts b,d,f,h,j- I think there are too many lines on one graph, it's very difficult to make sense of. I think split these in two and consider using a more colorblind-friendly colorscheme.

Figure 7 Why are parts e and f plotted as m/z 4344 and not f4344? You do that with the mass fraction of OA vs SO4, I think it would be useful for the organic markers

Figure S5 Could you normalise the y axis so that the heights are also the f's ie f44, f43 etc.

---

## Referee Comment (RC2) · Anonymous Referee #2 · 12 Jun 2018

Tremblay et al. describe aerosol growth events observed during the summers of 2015 and 2016 at Eureka and Alert in the Canadian Arctic. Aerosol size distributions were measured at both sites, with over a month of Aerodyne AMS data for non-refractory PM1 composition at Eureka. This is a unique dataset, particularly given the paucity of observational data available in the rapidly changing Arctic. I provide suggestions below primarily to clarify the text. In addition, the authors reference the NASA MODIS imagery quite a bit in the text, yet they provide no images in the text or SI; since the influence of open water is key to the results of this work, I highly recommend adding representative images as a figure in the main text.

[Figure]

Major Comments: Page 1, Lines 23-24, Page 9, Lines 32-33, Page 11, Lines 31-32: Please define the size range measured, rather than just "smaller than 100 nm". It appears from Fig 7 that 60-100 nm is the size range measured. Please state as such to avoid misleading the reader since much of this paper focuses on <50 nm particles. Since organics are expected to primarily responsible for aerosol growth, one would expect that the mass of 60-100 nm would be dominated by organics.

Introduction: It would be useful to add a brief summary of previous Arctic growth event papers to give greater context for the reader.

SMPS, OPC, & AMS size comparisons: It is not clear whether the diameters have been adjusted (between mobility, optical, vacuum aerodynamic) for comparisons between these instruments. This is necessary. See DeCarlo et al 2004 (AS&T). Please also label mobility diameter appropriately throughout (e.g. Fig. 1, 3, 5).

Page 6, Line 8: Provide the exact number of events identified. Please also define how these events were identified, and how they were differentiated from local emissions.

Page 6, Lines 31-32: This references circulation patterns in a paper that is over 10 years old. What was the meteorology during the observations presented herein? Was the meteorology similar between the two summers? Were there ever periods when a growth event was observed at one location and not the other? IF so, could differences in circulation or weather (e.g. precipitation) explain this?

Page 7, 2nd paragraph: Please test for statistical significance to bolster these arguments.

Page 7, Line 21: This sentence mentions 28 events, but the previous page (line 8) references 40 events. Please explain or fix this discrepancy.

Page 8, Line 21: Provide the growth rates observed by Nieminen et al in parentheses here for easy comparison. Please also clarify that by "all the events", I believe you mean "all five events"?

Page 8, Line 31: Comment on sea ice vs open water in the area at this time.

Section 3.2: Does m/z 79 (MSA tracer) vs size show a pattern?

Figure 4: What time resolution was used to make this plot (was averaging done?)? Since this paper focuses on ultrafine particles, it would also be useful to add a section to this plot with the same categories, but only showing the <100 nm particle concentration binned.

Figure S1: Please add a note, with references, that this doesn't include the instrument inlet efficiencies so that the reader is not confused.

Minor Comments: Page 1, Line 20: Fix typo.

Page 1, Lines 25-26: Define 'larger particles' here for the reader not familiar with the AMS size range. Similarly, please define "m/z 44" here in terms that the non-AMS reader will understand.

Page 2, Line 12: It would seem appropriate to cite an older paper by Leck here

Page 2, Lines 15 & 25: Add references.

Page 3, Line 4: Fix sentence phrasing – "which is photo-mediated" doesn't describe the Canadian Arctic.

Sections 2.1 & 2.2: Please provide tubing diameters in metric, rather than English, units. Also, inner diameters would be more useful than outer diameters.

Page 3, Lines 25-28: Provide sampling inlet information, as was done for the Eureka sampling.

Section 2.2 and Page 9, Line 11: Provide the size range of the AMS.

Page 5, Line 21: Please clarify what is meant by "in a straight line". Upwind?

Page 7, Line 11: Please clarify "this year".

Page 7, Line 22-23: Is this the case for all 23 other events? Please clarify.

Page 7, Line 26: Are these references misplaced? The data shown in Table 1 is original to this work.

Page 9, Lines 7-8: Please clarify – did only one growth event (of the 5?) have a similar growth rate and back trajectory? Did other growth events, for which growth rates weren't calculated, have similar back trajectories? This would be useful to know.

Page 9, Line 13: By "later in the year", do you mean "later in the summer"?

Page 10, Line 10-11: This sentence is not related to this paragraph.

Page 12: Line 1: Please clarify the exact size range measured in this previous work.

Figure 2b: Are these daily averages? This needs to be clarified.

Figure 6: Please state the years included in this plot.

Figure 7a-d: These plots are missing legends, which impacts interpretation of the figure.

Figure S2: Please note the averaging used for this plot. Were the diameters adjusted?

---

## Referee Comment (RC3) · Anonymous Referee #3 · 2 Jul 2018

1st July 2018

The paper *Characterization of aerosol growth events over Ellesmere Island during summers of 2015 and 2016* by Samantha Tremblay et al. investigates formation of new particles and their growth in the Canadian Arctic providing valuable information about the chemical composition of Aitken mode particles in this region. The number of studies on fine particles in the Arctic is increasing but our understanding is far from being exhaustive and measurements are generally sparse and incomplete. Upon investigation of size resolved chemical composition the authors claim that for aerosol smaller than about 100 nm the growth is dominated by organics. This would certainly be a high impact finding if is proven to be true, but the data presented here are not convincing enough to support this hypothesis and some major integrations are needed. Concluding, the paper is nicely written and addresses an important topic but there is some more work that needs to be done before publication in ACP.

**Major comments**

- **Influence of meteorology on measurements performed at PEARL**: In the paper is said that "growth events (and presumably nucleation) are more commonly observed when the inversion is weaker or not present" however only vertical profiles during event days are analysed. The authors should look at vertical profiles during non-event days and compare them with those presented in the paper. For example it would be useful to plot the histogram shown in fig. 6 both for event and non-event days and check if the difference is significant or not. Moreover, for certain days, it seems like growth is happening with a relatively strong temperature inversion. I would suggest to check the diurnal profile of water vapour mixing ratio (I assume this to be measured at PEARL), this should show an increase if the airmass is coming from the boundary layer compared with the free troposphere. In case no evidences for boundary layer air contribution are found it would be interesting to check if the particle chemical composition during the growth is different from other events.

- **Methanesulfonic acid contribution**: Mass spectra for 2 aerosol growth events are compared with lab measurements of MSA particles to prove that there are organics contributing to the growth besides MSA. The relative differences and the small correlation between ambient and lab spectra provides a qualitative argument in favour of this hypothesis but the authors should try to quantify the contribution of MSA to the total organic mass. I'm aware of the fact that, due to the

low resolution of a Q-AMS, this could be tricky but there are previous studies [1] reporting MSA concentration with the same instrument (the same paper is also cited by *Tremblay et al.*), so with the available data it should be possible to calculate at least an upper bound on the organic mass fraction attributable to MSA.

- **Size resolved aerosol chemical composition**: In figure 7, the authors show chemical composition down to 10 nm, whereas the sizing calibration is done until 80 nm. This extrapolation is beyond AMS capabilities (it would be incredibly good if an AMS could really measure 10 nm particles) and needs to be corrected, a non-AMS user could easily be fooled by these figures.

- **Aerosol composition during growth events**: the high organic-to-sulfate ratio reported in this paper for particles below 100 nm is surprising, in particular considering the marine sources, unfortunately the authors provide only a *snapshot* for 2 events while a more complete analysis should be performed to support their findings. The first thing that is not clear from figure 7 is why sulfate is present only in the larger particles, is this due to a second mode of aerosols that have a different history? The authors should comment on this. There are no indications about the period considered for the calculation of the size resolved chemical composition: is this an average over the whole growth event or a selection of a defined period? I would appreciate if the authors could show the size resolved chemical composition at 2/3 different stages of the event, this could tell a little bit more about the mechanisms beyond the growth. Moreover, it would be really useful if the authors could plot the total organic and sulfate time series concentration during the growth event. If the growth is mostly due to organics then I would expect to see an increase of organic concentration whereas sulfate should stay more or less constant. In addition, uncertainties on aerosol composition should be estimated and added to figure 7.

Finally, the authors conclude saying that "particles smaller than 100 nm in diameter are predominately organic with the organic-to-sulphate ratio increasing for smaller particle sizes", this is a general conclusion but is based on the analysis of only 2 events that does not provide any statistical basis for such kind of conclusion. For this reason the authors should look at the size resolved organic-to-sulfate ratio for all the events to check whether this statement is verified or not. It would be useful also to compare the size resolved chemical composition with non event days.

**Minor comments**

- The authors often speak of particle nucleation, I would avoid using this terminology in the paper because there are no measurement for particles below 10 nm. For this reason there are no proof that nucleation is really happening at the measuring sites (in particular considering the small growth rates reported in this paper).

- I would suggest to add a couple of sentences about iodine nucleation in the introduction. This has been proven to be a very effective mechanism in certain coastal regions [2] and a recent paper in the Arctic also showed evidence of this [3]. Later in the manuscript (chapter 3.2) the authors should also mention whether they see or not any evidence for iodine particles in their spectra.

- In chapter 3.1.2 growth rates for 5 selected events are reported. However, there is no mention to the method used to estimate the growth, nor to the size range considered for the calculation. The authors should add this information to the manuscript that is really important in particular when comparing with results from other studies.

- In chapter 3.2 a detailed analysis of fragments m/z 43 and 44 is provided, I wonder whether the authors can exclude any contamination from combustion or other sources (e.g. the generator) on fragment m/z 43. Moreover, I agree with referee #1 in saying that figure 8 doesn't show any clear trend and the authors should reconsider their conclusions here.

- Figure 3 shows a nice agreement between the PM1 as measured by the AMS and SMPS+OPC, however the authors should mention which density values were used to calculate the total mass for this comparison.

- Figure 4 reports the total particle number concentration at the 2 measurement sites, I would suggest to add a second box and whisker plot showing only the concentration of Aitken mode particles.

**References**

[1]   Lisa Phinney et al. "Characterization of the aerosol over the sub-arctic north east Pacific Ocean". In: *Deep Sea Research Part II: Topical Studies in Oceanography* 53.20 (2006). Canadian SOLAS: Subarctic Ecosystem Response to Iron Enrichment (SERIES), pp. 2410 –2433. ISSN: 0967-0645. DOI: https://doi.org/10.1016/j.dsr2.2006.05.044. URL: http://www.sciencedirect.com/science/article/pii/S0967064506002025 (cit. on p. 2).

[2]  Mikko Sipilä et al. "Molecular-scale evidence of aerosol particle formation via sequential addition of HIO3". In: *Nature* 537.7621 (2016), pp. 4–6. ISSN: 0028-0836. DOI: `10.1038/nature19314`. URL: `http://www.nature.com/doifinder/10.1038/nature19314` (cit. on p. 3).

[3]  J. D. Allan et al. "Iodine observed in new particle formation events in the Arctic atmosphere during ACCACIA". In: *Atmospheric Chemistry and Physics* 15.10 (2015), pp. 5599–5609. ISSN: 16807324. DOI: `10.5194/acp-15-5599-2015` (cit. on p. 3).

---

## Author Response (AR1)

**Key**

Black = Reviewer Comments, **Solid Blue = Responses**, *Italicized Blue = Modified Text*

We would like to thank the reviewers for taking the time to review our manuscript and for their thoughtful comments. Their feedback has helped us clarify and improve the manuscript. We have reproduced the reviewer comments in black text. For ease of review, our responses are given in blue text, while text that has been modified in the manuscript is quoted using blue italics. We would also like to point out that the numbering of the figures from the revised manuscript is used here in the responses.

**Referee 1 Comments**

**Main comments**

**RC1.1**

One thing to consider for good measure is iodine-based nucleation which has been observed in Arctic marine summers before. You should mention it in your introduction and then probably rule it out for your measurements.

**The text has been updated in the introduction with:**

*Furthermore, iodine may be important for particle nucleation in the Arctic (Mahajan et al. 2010; Allan et al. 2015; Raso et al. 2017), although the processes leading to either nucleation or particle growth are not necessarily the same.*

**And in the Section 3.2 with:**

*While iodine may contribute to particle nucleation, the low resolution of the quadrupole AMS prevents quantification of iodine. Given the low signal at m/z 127 in this study during growth events, we believe that iodine is at most a minor contributor of mass to Aitken mode particles.*

**RC1.2**

P3 paragraph 2

Why such a focus on GEOS-Chem? What about other models? Or just models in general?

**As already discussed near the end of the in the introduction similar discrepancies have been observed in the GLOMAP model. We have focused on GEOS-Chem because to our knowledge there has been more recent work on particle nucleation in the Arctic, especially the Canadian Arctic, using this model.**

**RC1.3**

Section 2.1

A map would be really useful here, showing where all the field sites are and preferably the orography as well

**The Figure below has been added to the manuscript in the supporting information.**

(a)

[Figure]

(b)

[Figure]

**Figure S2.** Local map of PEARL RidgeLab, SAFIRE and Eureka weather station (a) and zoomed-out map of Alert and Eureka (b).

**RC1.4**

P4 Paragraph 2

Could you move all the discussion of the tubing etc to the supplement? It's good to be thorough but this doesn't contribute to the science of the paper. I'm sure you'll agree it's also not the most interesting paragraph ever written...

**This portion of the text has been moved to the supporting information.**

**RC1.5**

P4 L23

Any idea why the difference in the OPC performance? How does that affect your AMS comparison?

**It is difficult to provide a conclusive reason for the small difference in OPC performance between 2015 and 2016. A possible explanation is that the size range compared isn't exactly the same for the two instruments, 300 – 487 nm for the SMPS versus 300 – 500 nm for the OPC. Therefore, there may be some differences in the actual particle size distribution for 2015 compared to 2016 that leads to a larger difference between the two instruments in 2015. It should be noted that the 2015 data also corresponds to only a portion of the summer and starts on 26 July 2015, whereas the 2016 data includes the full summertime period beginning on 16 June 2016 until 26 September 2016.**

**Overall, the comparison against the AMS $PM_1$ is excellent with a linear regression analysis yielding a slope of 1.16 and a correlation coefficient of 0.89, which is well within published uncertainty for AMS measurements (Middlebrook et al. Aerosol Sci. Technol., 46, 258-271, 2012). Therefore, the difference in OPC performance is not large enough to have a significant impact on the AMS comparison.**

**RC1.6**

P7 L25

This is written strangely, it sounds like you are saying you got the numbers from table 1 from the 3 papers you reference, which isn't the case. I think it would be better something like "Aerosol growth rates were calculated [say here very briefly how they are calculated and give the relevant reference(s)]. The initial growth rates from this study are included in Table 1." It would be useful to comment on the meaning of this average growth rate as well, since it takes a lot more mass to go from 100nm to 101 than to go from 10 to 11. Is the number skewed towards being more representative of growth at any particular part of the size distribution?

**We have clarified this section of the text and the updated version is copied below.**

*Initial aerosol growth rates were calculated following previously published methods (Kulmala et al. 2004; Hussein et al. 2005; Salma et al. 2011). Briefly, the SMPS size distributions were fitted with a multi-mode log-normal distribution, and then a linear regression analysis was performed on the geometric mean of the Aitken mode as a function of time for particle diameters between 10 – 30 nm. The initial growth rates calculated for this study are given in Table 1.*

**The influence of the absolute particle size on the growth rate, as mentioned by the reviewer, is limited by calculating the growth rate for only the smallest particle sizes. We have added some discussion of this consideration in Section 3.1.2.**

*It should be noted that the size range used for calculating growth rates in our work (10 – 30 nm) is slightly different from that of Collins et al. (4 – 20 nm) and Nieminen et al. (10 – 25 nm), which may contribute to our relatively slower growth rates.*

**RC1.7**

P8 L10 and Fig 6

Can you add on the inversion strength for non-nucleation events and see if there is an obvious difference? That would strengthen your hypothesis that it's the lack of inversion that's creating conditions conducive to nucleation at the measurement altitude

**We have made several modifications to the text to address this comment and to add analysis and discussion of non-nucleation (or more precisely the non-growth) events. Specifically, the second paragraph of section 3.1.2 and Figure 7 (in the new text) have been updated. In addition, we have added Table S2 and Figures S5 and S6 to the supporting information. The new text, tables and figures are included directly below in this response.**

*Furthermore, we conducted a similar analysis for six periods when particle concentrations were low and for six periods with a persistent accumulation mode (summarized in Table S2 and Figures S5 and S6). Figure 7 shows that the average inversion temperature during the growth events (0.3 ± 0.7 º C) was very similar to that during the selected periods with low particle concentration (0.3 ± 0.2 º C), whereas the average inversion temperature during periods with a persistent accumulation mode and elevated particle concentrations was much higher (2.5 ± 1.2 º C). The results shown in Figure 7 imply that growth events occur at the PEARL RidgeLab when the inversion is weak because, firstly, the low particle surface area and corresponding condensation sink in the marine boundary layer air allowed particle nucleation to occur, and secondly, the site was possibly influenced by more recent surface emissions that were less photochemically aged compared to air aloft. In contrast, when the inversion was strong, the aerosol and aerosol precursor species were more chemically aged due to slower transport into the free troposphere and thus the existing particles had already grown to*

*sizes corresponding to the accumulation mode. A few growth events were observed when the temperature inversion was larger, which may be due to the fact that the radiosondes were launched at the Eureka Weather Station located 11 km to the southwest of the PEARL RidgeLab. Thus, the temperature profile measured by a radiosonde may not be fully representative of that at the RidgeLab. Generally speaking, the observations reported here are consistent with previous work (Willis et al. 2016; Collins et al. 2017) suggesting that similar events measured in the Canadian Arctic are attributable to marine sources.*

[Figure]

**Figure 7.** Histogram of the number of growth events near Eureka, binned by the average change in the temperature from 10 to 600 m above sea level. The average temperature change of each event is provided in Table S1, and was calculated from radiosonde measurements during 2015 and 2016, as shown in Figure 6. The dashed line indicates the average change in temperature during periods of low particle concentrations as shown in Table S2 and Figure S5 (0.3 ± 0.2 °C), and the dotted line indicates the average change in temperature during periods with a persistent accumulation mode as shown in Table S2 and Figure S6 (2.5 ± 1.2 °C). The values in parenthesis are the averages and their standard deviations.

**Table S2.** List of selected periods of low and high particle concentrations observed near Eureka during the summers of 2015 and 2016 . The periods of high concentrations do not exhibit particle growth and are therefore distinct from the growth events. The SMPS measurements for the periods of low and high concentrations are shown in Figure S5 and S6, respectively.

| Non-Event (NE) | Time period (UTC) | |
| --- | --- | --- |
| | Start | End |
| NEA (low) | 2015-08-06  06:00 | 2015-08-07  02:00 |
| NEB (low) | 2015-08-24  11:00 | 2015-08-26  12:00 |
| NEC (low) | 2015-09-04  12:00 | 2015-09-05  18:00 |
| NED (low) | 2016-06-23  08:00 | 2016-06-24  15:00 |
| NEE (low) | 2016-06-30  12:00 | 2016-07-01  15:00 |
| NEF (low) | 2016-08-02  00:00 | 2016-08-02  12:00 |
| NEG (high) | 2015-08-10  07:00 | 2015-08-11  10:00 |
| NEH (high) | 2015-08-19  22:00 | 2015-08-24  10:00 |
| NEI (high) | 2015-08-29  07:00 | 2015-09-04  12:00 |
| NEJ (high) | 2016-07-13  19:00 | 2016-07-19  12:00 |
| NEK (high) | 2016-07-25  18:00 | 2016-07-26  03:00 |
| NEL (high) | 2016-07-30  00:00 | 2016-08-02  00:00 |

[Figure]

**Figure S5.** SMPS measurements of selected periods with low particle concentrations observed near Eureka during the summers of 2015 and 2016 as summarized in Table S2. Note that the figures display an additional 2 hours before and after the analyzed period. The sizes are mobility diameters measured by an SMPS, which are equal to the physical diameters under the assumption that the particles are spherical and contain no voids.

[Figure]

**Figure S6.** SMPS measurements of selected periods with high particle concentrations and without growth observed near Eureka during the summers of 2015 and 2016 as summarized in Table S2. Note that the figures display an additional 2 hours before and after the analyzed period. The sizes are mobility diameters measured by an SMPS, which are equal to the physical diameters under the assumption that the particles are spherical and contain no voids.

**RC1.8**

P10 L19

I'm not sure I agree there's a clear trend- there's a slight trend if you take some pretty big averages but mostly there's just random noise. The average data points are not very far apart either. How does that compare to what you expect in this environment? I think adding some literature values to Fig 8 would be useful to add some context.

**The first panel, panel (a), does show a distinct change in the $f_{44}$:$f_{43}$ graph during GE3. The trend is not as large compared to typical presentations of the $f_{44}$:$f_{43}$ space, because of the greater limits in our figure and relatively smaller changes in $f_{44}$ and $f_{43}$. The trend is less clear for the other event, GE6, and we have adjusted the text in the manuscript to more clearly explain this point. To our knowledge, only one other paper has reported growth events in this space and over these timescales. In that sub-tropical urban location, both the $f_{44}$ and $f_{43}$ were found to decrease as growth events evolved (Salimi et al. Atmospheric Chemistry and Physics, 15, 13475-13485, 2015). This contrasting behavior is interesting and may be attributable to fact that this previous work was performed in a location where other sources of particles besides nucleation (e.g. primary combustion particles) may have been contributing to the measured $f_{44}$ and $f_{43}$. In general, the $f_{44}$:$f_{43}$ values observed are consistent with values of oxygenated organic aerosols (a proxy for SOA) as measured at continental and marine locations (Ng et al. Atmospheric Chemistry and Physics, 10, 4625-4641, 2010; Choi et al. Atmospheric Environment, 171, 165-172, 2017). The text in Section 3.2 now reads:**

*For GE 3, there was a clear trend during the evolution of the growth event wherein both $f_{44}$ and $f_{43}$ increase, which is consistent with an increase in the relative concentration of carboxylic acids and non-acid oxygenates in the organic aerosol. For GE 6, a change in organic composition is not apparent. The $f_{44}$ may have increased, but the trend with time is more ambiguous than for GE 3. In comparison, the size-resolved composition measurements shown in Figure 8, show smaller particles have a higher fraction of m/z 43 and larger particles formed later during the growth events have a higher fraction of m/z 44 suggesting that aerosol growth led to an increase in the amount of oxidation due to, in part, the production of carboxylic acids. However, we emphasize that our results are for a very limited data set and further analysis of SOA composition during additional growth events using $f_{44}$ and $f_{43}$ would be necessary to confirm our observations.*

**RC1.9**

P11 L6

Could you put the NASA worldview images in the supplement?

**The Figure below has been added to the manuscript in the supporting information.**

**(a)**

[Figure]

**(b)**

[Figure]

**Figure S9.** Image of the ice coverage around Eureka during 25 June 2016 (a) and during 7 July 2016 (b) given by NASA Worldview.

**RC1.10**

Finally, I think it would be useful to discuss the implications for of your findings for CCN concentrations in the Arctic.

**This is an excellent suggestion and the following text has now been added to the conclusion section.**

*Previous work in the summer time Arctic found that particles smaller than 50 nm could be contributing to cloud droplet activation (Leaitch et al. 2016). The growth of small particles to diameters larger than 60 nm observed in our study could therefore make them an important contributor to CCN, ultimately impacting the radiation balance and hydrologic cycle.*

**Technical/stylistic corrections**

**RC1.11**

I found the tenses very confusing throughout. Take for example the abstract. You say "Measurements...were taken during…2015 and 2016". You then switch to the present (incorrectly in my view) and say "These events are observed beginning in June…marine sources are the primary cause." It's not like you took data from 2 decades and can really talk in general terms about events that you expect to happen every summer. You have data from 2 years, so it's appropriate to refer to your measurements and things that you observed in the past. Then you talk about what the graphs show in the present. Other examples of incorrect tense are P2L16, P3L24, P4L18, but there are many more. Please be consistent and refer to things that happened in the past, in the past tense.

**The manuscript has been revised to make the tenses consistent. In the interest of clarity, we did not highlight every change that was made to address this comment.**

**RC1.12**

P7 L23

Preceding not proceeding

**The text has been corrected.**

**RC1.13**

Figure 5 parts b,d,f,h,j

I think there are too many lines on one graph, it's very difficult to make sense of. I think split these in two and consider using a more colorblind-friendly colorscheme.

**The figure in question has been changed in the manuscript. In particular, we have split the old Figure 5 into two new figures (Figures 5 and 6 in the revised manuscript). We have also added patterned lines to the graphs of the temperature profiles. In total, we think these changes make the data and graphs easier to read.**

[Figure]

**Figure 5.** Five selected growth events near Eureka during the summers of 2015 and 2016. The grey dashed line indicates the start of each growth event. The sizes are mobility diameters measured by an SMPS, which are equal to the physical diameters under the assumption that the particles are spherical and contain no voids.

[Figure]

**Figure 6.** Vertical temperature profiles every 12 h during the 5 selected growth events that are shown in the previous figure. Data includes 24 h and 12 h prior to the start of the growth event.

**RC1.14**

Figure 7

Why are parts e and f plotted as m/z 43 44 and not f43 44? You do that with the mass fraction of OA vs SO4, I think it would be useful for the organic markers

**The Figure below has been added to the manuscript in the supporting information.**

[Figure]

**Figure S12.** Organic aerosol fraction measured at m/z 43 and m/z 44 during GE 3 (a) and GE 6 (b) near Eureka.

**RC1.15**

Figure S5

Could you normalise the y axis so that the heights are also the f's ie f44, f43 etc.

**The figure has been updated as shown below.**

[Figure]

**Figure S11.** AMS average ambient aerosol mass spectrum of GE 3 (a) and GE 6 (b) compared with the mass spectrum of MSA (c). The Ion Rate Fraction is the normalized Ion Rate (in Hz).

**Referee 2 Comments**

**RC2.1**

The authors reference the NASA MODIS imagery quite a bit in the text, yet they provide no images in the text or SI; since the influence of open water is key to the results of this work, I highly recommend adding representative images as a figure in the main text.

**We kindly refer the reviewer to our response to comment RC1.9.**

**Major Comments**

**RC2.2**

Page 1, Lines 23-24, Page 9, Lines 32-33, Page 11, Lines 31-32

Please define the size range measured, rather than just "smaller than 100 nm". It appears from Fig 7 that 60-100 nm is the size range measured. Please state as such to avoid misleading the reader since much of this paper focuses on <50 nm particles. Since organics are expected to primarily responsible for aerosol growth, one would expect that the mass of 60-100 nm would be dominated by organics.

**In all the different places specified by the reviewer the text has been changed to** "*between 50 and 80 nm*", **when before it was written** "*smaller than 100 nm*" **or** "*less than 100 nm*". **Please note that we have used a slightly different diameter range than suggested by the reviewer because we have converted the vacuum aerodynamic diameters to physical diameters.**

**RC2.3**

Introduction

It would be useful to add a brief summary of previous Arctic growth event papers to give greater context for the reader.

**The second and third paragraphs of the introduction have been restructured and additional references included to provide better context. The update text is quoted directly below.**

*In tropical marine locations, new particle formation tends to occur in the upper part of the troposphere, usually at the outflow of clouds, and these particles are entrained to the surface through mixing, which contributes to relatively stable aerosol size distributions (Hoppel et al. 1986; Clarke et al. 2006). In contrast, modelling studies of the Arctic summer show that persistent cloud and drizzle causes wet deposition and results in low condensation sinks at the surface (Browse et al. 2014; Croft et al. 2016a). These same studies show that these conditions can favour particle nucleation followed by growth between drizzle events. This is supported by surface observations of aerosol size distributions in the Arctic at Alert and Ny-Ålesund that show an annual cycle during which summertime surface aerosols exhibit much smaller particle diameter than wintertime aerosols (Tunved et al. 2013; Croft et al. 2016a). Additional surface observations have suggested that new particle formation could be the source of these small particles, with dimethyl sulphide (DMS) emitted from the ocean being a key gaseous precursor of less volatile species, such as sulphuric acid and methanesulphonic acid, that contributes to aerosol mass (Chang et al. 2011; Leaitch et al. 2013).*

*Sulphuric acid has long been known to contribute to new particle formation and growth events (Twomey 1977; Charlson et al. 1992; Napari et al. 2002; Lohmann and Feichter 2005; Kirkby et al. 2011; Almeida et al. 2013; Croft et al. 2016b). More recent work has shown that in coastal Arctic environments, ammonia from sea-bird colonies can contribute to new particle formation (Croft et al. 2016b). In addition, organic compounds, especially those with lower volatilities, have also been found to contribute secondary aerosol mass to particle growth and nucleate new particles in forested and anthropogenically influenced regions (Allen et al. 2000; Zhang et al. 2009; Pierce et al. 2012; Riipinen et al. 2012) as well as in laboratory studies (Kirkby et al. 2016; Trostl et al. 2016). Box models have inferred the contribution of non-sulphur species (i.e. organic compounds) to aerosol growth in Greenland (Ziemba et al. 2010) and in tropical marine cloud outflow (Clarke et al. 1998). Burkart et al. (2017) provided indirect evidence that organic compounds contribute to aerosol growth in high-latitude marine environments using both microphysical modeling of a particle growth event as well as cloud condensation nuclei (CCN) hygroscopicity measurements. In a comparison of ship-borne observations in the Canadian Arctic in 2014 and 2016, Collins et al. (2017) found that increased activity in marine microbial communities along with greater solar radiation and lower sea ice concentrations contributed to new particle formation and growth. Recent work by Mungall et al. (2017) in the Canadian Arctic also suggests that a photo-mediated marine source of oxygenated volatile organic compounds could produce precursor vapors for new particle formation or growth. Furthermore, iodine may be important for particle nucleation in the Arctic (Mahajan et al. 2010; Allan et al. 2015; Raso et al. 2017), although the processes leading to either nucleation or particle growth are not necessarily the same.*

**RC2.4**

SMPS, OPC, & AMS size comparisons

It is not clear whether the diameters have been adjusted (between mobility, optical, vacuum aerodynamic) for comparisons between these instruments. This is necessary. See DeCarlo et al 2004 (AS&T). Please also label mobility diameter appropriately throughout (e.g. Fig. 1, 3, 5).

**Following DeCarlo et al. AS&T 2004, if it is assumed that the particles are spherical and contain no voids, then their mobility diameter will be equal to their physical (or geometric) diameter. The assumed particle shape and morphology are reasonable given the evidence that the largest portion of the particles mass is due to secondary formation (either secondary organic aerosols or sulphate).**

**To address this point we have modified the text in Section 2.1.**

*Following the work of DeCarlo et al. (2004), the mobility diameter measured by the SMPS was assumed to be equal to the physical diameter, which would be valid if the sampled particles are spherical and contain no voids. This is a reasonable assumption given the secondary origin of the observed particles.*

**We have also added the following sentence to the figure captions for those figures containing SMPS data. We prefer to not write "mobility diameter" directly on the figure axes as that would create a lot of visual clutter.**

*The sizes are mobility diameters measured by an SMPS, which are equal to the physical diameters under the assumption that the particles were spherical and contained no voids.*

**The vacuum aerodynamic particle diameter measured by the AMS is now adjusted to be mobility diameter and thus is now equal to the physical diameter, under the assumptions given above. We have added the following text to the manuscript in Section 2.2 to explain the adjustment.**

*Vacuum aerodynamic diameters measured by the AMS were converted to physical diameter under the assumption that the particles were spherical, contained no voids, and had a density of 1.25 g cm$^{-3}$ (DeCarlo et al. 2004). This density is typical for ambient organic aerosol (Middlebrook et al. 2012), and was selected for this study since the analysis was focused on the particle composition during the predominantly organic aerosol growth events.*

**Lastly, the OPC diameter should be equal to physical diameter insofar as the Mie scattering curve of the calibration particles is similar to that of the actual ambient aerosol. In this case, polystyrene latex spheres were used by the manufacturer for calibration. While the Mie Scattering curve for ambient aerosol may vary from that of the PSLs, any variation will be small and likely within specified instrument size accuracy of ±10%, insofar as the ambient particles are spherical and predominately organic (and therefore have an index of refraction similar to PSLs). These assumptions seem reasonable for this study. The following text was added to Section 2.1.**

*It was further assumed that the OPC diameter was equal to the physical diameter, given that the Mie scattering curve of the ambient aerosols was likely within 10% of that of the calibration particles composed of polystyrene latex spheres.*

**RC2.5**

Page 6, Line 8

Provide the exact number of events identified. Please also define how these events were identified, and how they were differentiated from local emissions.

**The text in the Section 3.1.1 has been updated with:**

*Figure 1 shows the aerosol size distributions measured at the PEARL RidgeLab and at Alert for 16 June to 26 September 2016. Particle growth events were evident at both sites. In total, 34 events with elevated concentrations of small particles (< 20 nm diameter) were observed at the PEARL RidgeLab during this period, 22 of which were followed by growth lasting between 2 to 6 days. It is important to note that the local anthropogenic emissions should be completely negligible due to the extremely remote position of the site. The electricity for the PEARL RidgeLab is generated by a small power plant located 11 km from the site and there is no indication from the measurements that the site is significantly influenced by emissions from the power plant or the Eureka Weather Station.*

[Figure]

**To confirm that local emissions did not influence the site and the observed growth events, the figure above shows the SMPS data during August and September 2016 (a), when a PAX (Photoacoustic Extinctiometer) was installed at the PEARL RidgeLab. The PAX measures the aerosol light scattering and absorption, as well as the concentration of black carbon (BC). The PAX measurements for this period (b) show that growth events occurred in the absence of BC during the end of the summer 2016. Furthermore, during the summer of 2017 (c), the concentration of BC was extremely low and consistent with background arctic conditions (Law and Stohl, Science, 315, 1537-1540, 2007). From these observations, it is possible to conclude that there was no significant contribution from local emissions given the very low concentrations of BC at the PEARL RidgeLab.**

**RC2.6**

Page 6, Lines 31-32

This references circulation patterns in a paper that is over 10 years old. What was the meteorology during the observations presented herein? Was the meteorology similar between the two summers? Were there ever periods when a growth event was observed at one location and not the other? IF so, could differences in circulation or weather (e.g. precipitation) explain this?

**We agree that the reference may not accurately describe circulation patterns during our study period, and thus it has been deleted. Available observations of temperature, RH and wind speed at the PEARL RidgeLab and Alert for the relevant periods in 2015 and 2016 are shown below. They are surprisingly similar, suggesting that the meteorology experienced at both sites was similar. Nevertheless, growth events were sometimes observed at one site but not the other. For some of these single site growth events, it is possible that differences in meteorology, especially RH, could be the cause of the discrepancy. However, it is not consistently different, with some single site growth events occurring when the RH is the same and some simultaneous growth events occurring when the RH is different. As such, we do not believe we can attribute specific differences in weather to the observed differences in growth events. This is consistent with sites at lower latitudes where identical meteorological conditions do not always lead to nucleation and growth events (e.g. Jeong et al. Atmospheric Chemistry and Physics, 10, 7979-7995, 2010.)**

[Figure]

**RC2.7**

Page 7, 2nd paragraph

Please test for statistical significance to bolster these arguments.

**This paragraph has been shortened so that the arguments are more precise.   (See the copied text below.)**

*In order to evaluate the influence of the appearance of small particles and growth events on the particle number concentrations at the two sites, the total concentration and the concentration of particles with a size between 10 to 100 nm, measured by the SMPS are summarized in Figure 4 for 27 Jul – 9 Sep 2015 and 2016. The particle concentrations are similar at both sites and for both periods. The one exception is that the 90th percentile was higher for Alert in 2015, which was driven by two events with especially elevated particle concentrations. Coinciding events were observed at Eureka, but the particles concentrations were much lower. The reason for the elevated concentrations at Alert but not at Eureka is unknown. It is important to note that for 2016, the mean is approximately 50 – 100 particles cm⁻³ higher than the results shown in Figure 4 if data from 16 Jun – 26 Sep 2016 are analyzed instead. This can be explained by the fact that the total duration of growth events was longer in June and July compared to events occurring in August and September.*

**For reference, the mean and the standard deviation for the particle concentrations at each site and for 2015 and 2016 are given below, which further illustrates the similarity in the frequency distribution of the particle number concentration. The data in the table is for particle diameters between 10 – 487 nm, and very similar results are obtained when limiting the size range to 10 – 100 nm.**

| | Mean ± Standard Deviation (particles/cm³) |
|---|---|
| **Alert 2015** | 204 ± 290 |
| **Alert 2016** | 168 ± 125 |
| **Eureka 2015** | 145 ± 170 |
| **Eureka 2016** | 200 ± 127 |

**RC2.8**

Page 7, Line 21

This sentence mentions 28 events, but the previous page (line 8) references 40 events. Please explain or fix this discrepancy.

**The text has been clarified regarding the number of events. The text on page 6 now reads as follows.**

*Figure 1 shows the aerosol size distributions measured at the PEARL RidgeLab and at Alert for 16 June to 26 September 2016. Particle growth events were evident at both sites. In total, 34 events with elevated concentrations of small particles (< 20 nm diameter) were observed at the PEARL RidgeLab during this period, 22 of which were followed by growth lasting between 2 to 6 days.*

**In Section 3.1.2, the new text is as follows.**

*To further analyze the growth events and periods with elevated concentrations of ultrafine particles, two different sets of case studies were selected comprising 5 (Table 1) and 28 events (Table S1). The latter represents all the growth events observed during the measurement period (22 events during 2016 and 6 events during the shorter 2015 period), and the smaller set of 5 was used to calculate growth rates. This subset was chosen because they were distinct, without overlap with preceding or subsequent growth events and exhibited relatively smooth growth curves.*

**In total, 28 growth events were observed during the entire study period, with 22 of those events occurring in 2016. The number of 40 given in the previous version of the manuscript was an estimated value and incorrect.**

RC2.9

Page 8, Line 21

Provide the growth rates observed by Nieminen et al in parentheses here for easy comparison. Please also clarify that by "all the events", I believe you mean "all five events"?

**The text has been updated with:**

*Previous studies have characterized aerosol growth rates in remote regions including the Arctic. In particular, Collins et al. (2017) reported growth rates ranging from 0.2 – 15.3 nm h$^{-1}$ during two research cruises conducted in the Canadian Arctic and calculated a corresponding average growth rate of 4.3 ± 4.1 nm h$^{-1}$. Similarly, Nieminen et al. (2018) reported for Alert and Mt Zeppelin, Norway, that the average growth rates, between June and August, were 1.1 and 1.2 nm h$^{-1}$, respectively for the years 2012-2014 and 2005-2013. In our study, growth rates ranged from 0.1 – 1.0 nm h$^{-1}$ for the aerosols at the PEARL RidgeLab and at Alert, with an average rate of 0.5 ± 0.3 nm hr$^{-1}$ (Table 1). These values are consistent with those reported in Collins et al. and in Nieminen et al. It should be noted that the size range used for calculating growth rates in our work (10 – 30 nm) is slightly different from that of Collins et al. (4 – 20 nm) and Nieminen et al. (10 – 25 nm), which may contribute to our relatively slower growth rates. Lastly, the growth rates are similar for all 5 events analyzed in Table 1, which suggests that the atmospheric processes (e.g. the condensation of semi-volatile or low volatility vapors to the particle surfaces as discussed below) and conditions governing the growth events are similar for all the events in this study.*

**RC2.10**

Page 8, Line 31

Comment on sea ice vs open water in the area at this time.

**We now include NASA Worldview images in Figure S13, where open waters south of Ellesemere Island can be seen starting on 25 June 2016 and near Eureka Sound starting on 7 July 2016.**

**RC2.11**

Section 3.2

Does m/z 79 (MSA tracer) vs size show a pattern?

[Figure]

**The figure above shows the m/z 79 size distribution for GE 3 (a) and GE 6 (b). The m/z 79 size distribution does not resemble the m/z 43 or the m/z 44 distribution. It is also different from the total organic size distribution. Since we do not have high mass resolution, it is possible there are other organic fragments contributing to m/z 79. Nevertheless, MSA does not appear to be an important contributor to the total mass, because if that were the case the m/z 79 distribution and the total organic distribution would be similar. The following text has been added to this section:**

*The size distribution of m/z 79 also peaks at larger sizes, suggesting that any MSA present would be in the accumulation mode*

**RC2.12**

Figure 4

What time resolution was used to make this plot (was averaging done?)? Since this paper focuses on ultrafine particles, it would also be useful to add a section to this plot with the same categories, but only showing the <100 nm particle concentration binned.

**The figure has been updated to:**

[Figure]

**Figure 4.** Box and whiskers plots of hourly particle number concentrations for diameters between **(a)** 10 – 487 nm, and **(b)** 10-100 nm measured during 27 Jul to 9 Sep 2015 and 2016 in Alert and at the PEARL RidgeLab near Eureka. The plots indicate the 10th, 25th, 50th, 75th and 90th percentiles.

**RC2.13**

Figure S1

Please add a note, with references, that this doesn't include the instrument inlet efficiencies so that the reader is not confused.

**The figure caption has been updated to read as follows.**

*Figure S1. Inlet particle transmission efficiency for the AMS (a) and SMPS (b) at the PEARL RigdeLab. Note that the curves do not include the instrument transmission efficiencies for the AMS (Jayne et al. Aerosol Sci. Technol. 2000, 33, 49-70) or for the SMPS (Wiedensohler et al. Atmos. Meas. Tech. 2012, 5, 657-685).*

**Minor Comments**

**RC2.14**

Page 1, Line 20

Fix typo.

**We have replaced "growths" with "growth"**

**RC2.15**

Page 1, Lines 25-26

Define 'larger particles' here for the reader not familiar with the AMS size range. Similarly, please define "m/z 44" here in terms that the non-AMS reader will understand.

**We have updated the text as indicated below.  The term "m/z 44" has been deleted from this section of the abstract as well.**

*The oxidation of the organics also changed with particle size, with the fraction of organic acids increasing with diameter from 80 to 400 nm.*

**RC2.16**

Page 2, Line 12

It would seem appropriate to cite an older paper by Leck here.

**Reference is now made to the following paper.**

**Leck and Bigg, Source and evolution of the marine aerosol—A new perspective, Geophysical Research Letters, 32, L19803, 2005.**

**RC2.17**

Page 2, Line 15 & 25

Add references.

**A reference has been added to Line 15 of the previous version (Croft et al. Processes controlling the annual cycle of Arctic aerosol number and size distributions, Atmospheric Chemistry and Physics, 16, 3665-3682, 2016) and the text and references have been re-organized to read as follows.**

*In contrast, modelling studies of the Arctic summer show that persistent cloud and drizzle causes wet deposition and results in low condensation sinks at the surface (Browse et al. 2014; Croft et al. 2016a). These same studies show that these conditions can favour particle nucleation followed by growth between drizzle events.*

**RC2.18**

Page 3, Line 4

Fix sentence phrasing – "which is photo-mediated" doesn't describe the Canadian Arctic.

**The sentence has been re-worded. The updated text is copied below for the reviewer's convenience.**

*Recent work by Mungall et al. (2017) in the Canadian Arctic also suggests that a photo-mediated marine source of oxygenated volatile organic compounds could produce precursor vapors for new particle formation or growth.*

**RC2.19**

Section 2.1 & 2.2

Please provide tubing diameter in metric, rather than English, units. Also, inner diameters would be more useful than outer diameters.

**The text has been updated with:**

*At the PEARL RidgeLab, the instruments sampled year-round through a common aerosol inlet, made of 6 m of stainless steel tubing with a 25.4 mm outer diameter (OD) and an inner diameter (ID) of 22 mm, sampling 2 m above the roof of the laboratory with a total flow rate of 11 L/min, as previously described by Kuhn et al. (2010). The SMPS flow passed first through 0.5 m of 25.4 mm OD and 22 mm ID stainless steel tubing*

*connected to the common aerosol inlet; this flow then entered a 9.53 mm OD stainless steel tube with a length of 0.45 m and finally passed through a 6.35 mm OD and 4.72 mm ID copper tube that was 1.05 m long. For the OPC, the flow passed from the common aerosol inlet into a 12.7 mm OD copper tube with an ID of 9.40 mm and a length of 0.8 m, and then into a 6.35 mm OD copper tube with an ID of 4.72 mm and a length of 0.58 m, which was connected to the OPC by 6.35 mm OD conductive rubber tubing with an ID of 3.18 mm and a length of 0.04 m. Particle transmission efficiency to the SMPS has been calculated and the resulting transmission curve is shown in Figure S1 (von der Weiden et al. 2009). From the common aerosol inlet, the AMS flow passed first through 0.5 m of 25.4 mm OD and 22 mm ID stainless steel tubing and then through a 9.53 mm OD stainless steel tube with a length of 0.45 m before entering the AMS.*

**And following the recommendation given in comment RC1.4, this portion of the text is now in the supporting information.**

**RC2.20**

Page 3, Lines 25-28

Provide sampling inlet information, as was done for the Eureka sampling.

**We have updated the text to reference the appropriate articles that contain the sampling inlet information.**

*Details of the aerosol sampling inlet at Alert are described in the previous work of Leaitch et al. (2013) and Leaitch et al. (2018).*

**RC2.21**

Section 2.2 and Page 9, Line 11

Provide the size range of the AMS.

**The text has been updated with:**

**In Section 2.2:**

*Both hourly bulk and size-resolved concentrations were measured by switching between mass spectrometry (MS) mode and particle time-of-flight (PToF) mode, which provides quantitative measurements in the range of 50 to 1000 nm (aerodynamic diameters).*

**And the Section 3.2:**

*AMS measurements of aerosol chemical composition and mass concentration for the summer of 2015 are shown in Figures 3a and 3b, where the $PM_1$ mass concentrations include the four dominant types of non-refractory aerosol.*

**RC2.22**

Page 5, Line 21

Please clarify what is meant by "in a straight line". Upwind?

**We agree with the reviewer that the term "in a straight line" is unnecessary and thus confusing. It has been deleted. Simply stated, the distance between the PEARL RidgeLab and the ECCC Weather Station is 11 km.**

**RC2.23**

Page 7, Line 11

Please clarify "this year".

**The text has been updated in Section 3.1.1 with:**

*The one exception is that the 90th percentile was higher for Alert in 2015, which was driven by two events with especially elevated particle concentrations. Coinciding events were observed at Eureka, but the particles concentrations were much lower. The reason for the elevated concentrations at Alert but not at Eureka is unknown.*

**RC2.24**

Page 7, Line 22-23

Is this the case for all 23 other events? Please clarify.

**The text has been updated with a new sentence in the Section 3.1.2:**

*To further analyze the growth events and periods with elevated concentrations of ultrafine particles, two different sets of case studies were selected comprising five (Table 1) or 28 events (Table S1). The latter represents all the growth events observed during the measurement period (22 events during 2016 and 6 events during the shorter 2015 period), and the smaller set of 5 was used to calculate growth rates. This subset of events was chosen because they were distinct, without overlap with proceeding or subsequent growth events and exhibited relatively smooth growth curves. The remaining 23 growth events were sometimes interrupted, presumably due to changes in air mass origin, or consisted of several events overlapping each other. All of the 5 growth events presented in Table 1 represent complete and smooth growth events that were suitable for calculating growth rate.*

**RC2.25**

Page 7, Line 26

Are these references misplaced? The data shown in Table 1 is original to this work.

**We kindly refer the reviewer to our response to comment RC1.6 for which we have reorganized the references and added a short description of the method for calculating the growth rate.**

**RC2.26**

Page 9, Lines 7-8

Please clarify – did only one growth event (of the 5?) have a similar growth rate and back trajectory? Did other growth events, for which growth rates weren't calculated, have similar back trajectories? This would be useful to know.

**We have added back-trajectories for additional growth events in Figure S7, and the text in Section 3.1.3 has been updated accordingly.**

*To understand the influence of the air mass history on the occurrence of the growth events, back-trajectories were calculated using FLEXPART (Figure S7) for Eureka and Alert. (The particle size distributions for the analyzed events for Eureka and Alert are shown in Figures 5 and S8, respectively). This calculation permits the precise evaluation of the spatial distribution of the potential emissions sensitivity at the beginning of each growth event. In general, these calculations show that the aerosols measured at the PEARL RidgeLab are mostly influenced by source regions located in the Canadian Arctic Archipelago, in Baffin Bay and to the north of Ellesmere Island. These results mostly coincide with the research reported by Collins et al. (2017), in which they observed high concentrations of ultrafine particles in these regions. Furthermore, the analyzed growth events generally have similar air mass histories for both the PEARL RidgeLab and Alert for a given event. The exception is GE 30 at the PEARL RidgeLab, which began on 25 June 2016, was more influenced by areas near and further north of Alert with a small contribution from the Nares Strait region. Interestingly, NASA Worldview images (https://worldview.earthdata.nasa.gov/) show that on 25 June 2016 (Figure S13) and for several preceding days, while the ocean in these regions was mostly covered in sea-ice, the potential emissions sensitivity was still influenced by large areas of open water. In conclusion, growth events can occur within air masses with different back-trajectories, as also reported by Collins et al. (2017), although the potential emissions sensitivities for the five growth events shown in Figure S7 have a substantial amount of overlap.*

[Figure]

**Figure S7.** Evaluation of the air mass history during the five selected growth events summarized in Table 1 and shown in Figure 5 of the main text. The back-trajectory and potential emissions sensitivity were calculated using FLEXPART. The left column corresponds to air masses arriving at Eureka and the right column corresponds to those arriving at Alert.

[Figure]

**Figure S7.** Evaluation of the air mass history during the five selected growth events summarized in Table 1 and shown in Figure 5 of the main text. The back-trajectory and potential emissions sensitivity were calculated using FLEXPART. The left column corresponds to air masses arriving at Eureka and the right column corresponds to those arriving at Alert (continued).

[Figure]

**Figure S8.** SMPS measurements of 4 growth events at Alert during the summers of 2015 and 2016 corresponding to the periods summarized in Table 1. The sizes are mobility diameters measured by an SMPS, which are equal to the physical diameters under the assumption that the particles were spherical and contained no voids.

**RC2.27**

Page 9, Line 13

By "later in the year", do you mean "later in the summer"?

**The text has been updated with:**

*In contrast, later in the summer (approximately 30 August 2015 to 5 September 2015) there is a period of larger particles when the mass concentration of sulphate is higher than the organic component.*

**RC2.28**

Page 10, Line 10-11

This sentence is not related to this paragraph.

**This sentence :** *"Overall, the size-resolved measurements indicate that for the two growth events analyzed here, which occurred during summer 2015, the mass of the growing particles is predominately organic matter."* **has been removed from the article.**

**RC2.29**

Page 12, Line 1

Please clarify the exact size range measured in this previous work.

**The text has been updated with:**

*This result is in contrast to previous indirect measurements of aerosol composition using a volatility tandem differential mobility analyzer system installed near Ny-Ålesund, Svalbard (Giamarelou et al. 2016), which suggested that 12 nm particles were predominately ammonium sulphate, although it was not possible in that study to conclusively distinguish ammonium sulphate from organics with similar volatility.*

**RC2.30**

Figure 2b

Are these daily averages? This needs to be clarified.

**The caption has been updated with:**

*Note that the data in the scatter plot correspond to daily averages of particles with diameters between 20 and 70 nm.*

**RC2.31**

Figure 6

Please state the years included in this plot.

**The caption has been updated with:**

*The average temperature change of each event is provided in Table S1, and was calculated from radiosonde measurements during 2015 and 2016, as shown in Figure 6.*

**RC2.32**

Figure 7a-d

These plots are missing legends, which impacts interpretation of the figure.

**We updated the figure to include the legend.**

**RC2.33**

Figure S2

Please note the averaging used for this plot. Were the diameters adjusted?

**Hourly averages were used for this plot, and this averaging interval is now indicated in the figure caption. The equivalence of the SMPS and OPC diameters is discussed above in our response to comment RC2.4, and we kindly refer the reviewer to that response.**

**Referee 3 Comments**

**Major comments**

**RC3.1**

*Influence of meteorology on measurements performed at PEARL:*

In the paper is said that "growth events (and presumably nucleation) are more commonly observed when the inversion is weaker or not present" however only vertical profiles during event days are analysed. The authors should look at vertical profiles during nonevent days and compare them with those presented in the paper. For example it would be useful to plot the histogram shown in fig. 6 both for event and non-event days and check if the difference is significant or not.

**We kindly refer the reviewer to our response to comment RC1.7.**

RC3.2

Moreover, for certain days, it seems like growth is happening with a relatively strong temperature inversion. I would suggest to check the diurnal profile of water vapour mixing ratio (I assume this to be measured at PEARL), this should show an increase if the airmass is coming from the boundary layer compared with the free troposphere. In case no evidences for boundary layer air contribution are found it would be interesting to check if the particle chemical composition during the growth is different from other events.

**Based on the reviewer's suggestion, we compared observations at the surface, as measured at the Eureka weather station, and at the PEARL RidgeLab. The figure below illustrates the water vapour mixing ratio at the two heights. Only data from 2016 are included because observations were missing from the PEARL RidgeLab for most of August 2015, when growth events were observed. The first box-and-whiskers plot represents all times when data were available between June and September 2016 inclusive, and the second (GE) represents values when the growth events were occurring in 2016. (Note that the whiskers represent the farthest point that is less than or equal to the interquartile difference multiplied by a factor of 1.5.) Based on the lack of a difference in the water vapour mixing ratio, it appears that the air is fairly well-mixed between the surface and the PEARL RidgeLab. In addition, box-and-whiskers plots are given for periods when a growth event was not observed and either low concentrations of aerosols were present (NE low) or a persistent accumulation mode was present (NE High). These selected "non-events" are summarized in Table S2 of the supporting information. The NE low plot is similar to the All and GE plots, but the NE High plot has surprisingly a negative median indicating that the airmass at the PEARL RidgeLab had a higher water vapour mixing ratio compared to the Eureka weather station. Overall, these results suggest that the PEARL RidgeLab is generally influenced by the surface throughout the summer, including during growth events. There were however periods of elevated concentrations of relatively large particles that were not growing (as shown in Figure S6), and the water vapour observations during these periods would be consistent with an airmass at the PEARL RidgeLab that is more influenced by regions warmer than the surface at Eureka. We note lastly that the NE High periods also tended to exhibit a more pronounced inversion as described in our response to comment to RC1.7.**

**Water Vapour Jun-Sep 2016**

**RC3.3**

*Methanesulfonic acid contribution:*

Mass spectra for 2 aerosol growth events are compared with lab measurements of MSA particles to prove that there are organics contributing to the growth besides MSA. The relative differences and the small correlation between ambient and lab spectra provides a qualitative argument in favour of this hypothesis but the authors should try to quantify the contribution of MSA to the total organic mass. I'm aware of the fact that, due to the low resolution of a Q-AMS, this could be tricky but there are previous studies [1] reporting MSA concentration with the same instrument (the same paper is also cited by Tremblay et al.), so with the available data it should be possible to calculate at least an upper bound on the organic mass fraction attributable to MSA.

**We have addressed this comment by adding the following text to the manuscript in section 3.2.**

*To further verify these findings, the AMS fragmentation table was also modified to separately quantify MSA following Phinney et al. (2006), but the concentration of MSA was generally at or below the detection limit (0.021 $\mu g/m^3$ as determined by multiplying by 3 the standard deviation when the AMS was sampling through a filter). Based on this detection limit, the MSA concentration was 5% or less of the total organic and sulfate mass concentration during the two measured growth events, GE3 and GE6.*

**RC3.4**

*Size resolved aerosol chemical composition:*

In figure 7, the authors show chemical composition down to 10 nm, whereas the sizing calibration is done until 80 nm. This extrapolation is beyond AMS capabilities (it would be incredibly good if an AMS could really measure 10 nm particles) and needs to be corrected, a non-AMS user could easily be fooled by these figures.

**It is true that the calibration is only performed to a vacuum aerodynamic diameter of 80 nm. However, the AMS transmission curve allows qualitative detection of particles with sizes down to approximately 30 nm vacuum aerodynamic diameter (Jayne et al. *Aerosol. Sci. Technol.* 2000, *33,* 49-70). Furthermore, it is also useful to display the size distribution trace to 10 nm in order to verify that the AMS PToF baseline is correctly located near zero. Nevertheless, we agree with the reviewer that particle size calibration must be extrapolated below 80 nm, which means that the AMS size distribution data should not be interpreted in a quantitative manner below this diameter. We have therefore added the following sentence to Section 2.2 of the manuscript.**

*Importantly, it should be noted that the extrapolation of the aerodynamic diameter calibration below 80 nm is not well constrained, so particle size data below this diameter should be considered qualitative rather than quantitative.*

**RC3.5**

*Aerosol composition during growth events:*

The high organic-to-sulfate ratio reported in this paper for particles below 100 nm is surprising, in particular considering the marine sources, unfortunately the authors provide only a snapshot for 2 events while a more complete analysis should be performed to support their findings.

**We agree with the reviewer that, ideally, a much longer measurement campaign would be carried out to characterize the aerosol chemistry. Unfortunately, it has continued to be logistically challenging and very expensive to conduct AMS measurement in this region, so there is no additional data available before or after the period shown in the article. To reflect this, we have updated the conclusions and text to emphasize that the conclusions are drawn from a limited data set. On the other hand, there is now an emerging body of literature, including this study, which supports the conclusion that secondary organics are an important contributor to aerosol growth in the Arctic.**

**The text has been updated in several different sections in response to this comment:**

**Abstract:**

*It was found that particles with diameters between 50 and 80 nm during these growth events were predominately organic with only a small sulphate contribution.*

**Introduction:**

*The mass spectrometry measurements indicate that these ultrafine particles (<100 nm) were predominately organic during the observed growth events. This work builds on other studies that have indirectly characterized the organic content of Aitken mode aerosols in the Arctic (Burkart et al. 2017) and have measured oxidised volatile organic compounds in the Arctic atmosphere (Mungall et al. 2017).*

**Conclusion:**

*Moreover, in this study AMS measurements showed that particles between 50 and 80 nm in diameter during two observed growth events were predominately organic.*

**RC3.6**

The first thing that is not clear from figure 7 is why sulfate is present only in the larger particles, is this due to a second mode of aerosols that have a different history? The authors should comment on this. There are no indications about the period considered for the calculation of the size resolved chemical composition: is this an average over the whole growth event or a selection of a defined period?

**The size resolved chemical composition was calculated by averaging over the whole growth event using the times indicated in Table 1 of the manuscript. This averaging is now mentioned in the Figure 8 caption.**

**We agree with the reviewer that the most likely explanation for the relatively high concentration of sulfate in the larger particles is the presence of a second aerosol mode with a different history. We have added a sentence to the text in Section 3.2 to address this point.**

*Between 80 and 1000 nm, the measured aerosol composition changes depending on the particle size and the larger aerosol particles contain a greater fraction of sulphate. The higher concentration of sulphate in the larger particles is most likely explained by the presence of a distinct accumulation mode having a history and source different from the Aitken mode aerosols.*

**RC3.7**

I would appreciate if the authors could show the size resolved chemical composition at 2/3 different stages of the event, this could tell a little bit more about the mechanisms beyond the growth.

The figure below shows the unnormalized organic and sulphate aerosol concentration for GE3 and GE6 for the first half (a and b) and the second half (c and d) of each event. We can see clearly the organic size distribution is shifting to larger sizes from the beginning to the end of the growth event. The unnormalized data has not been mathematically normalized to the bulk measured AMS concentration. In this case normalizing is not necessary because only particle sizes are compared. (The AMS diameter has been adjusted to match the SMPS diameter.)

[Figure]

**RC3.8**

Moreover, it would be really useful if the authors could plot the total organic and sulfate time series concentration during the growth event. If the growth is mostly due to organics then I would expect to see an increase of organic concentration whereas sulfate should stay more or less constant.

As seen in the figure below, the sulphate mass shows a much smaller increase compared to the organic mass at the beginning of the two growth events. At the end of GE 3, other processes begin to affect the aerosol size distribution, which results in a small overall loss in total aerosol mass (0.06 µg/m³). However, the expected

**increase in aerosol mass during GE 6 based on the SMPS data (0.12 – 0.36 μg/m³) is reflected in the time series of the organic concentration. This figure has been added to the manuscript in the supporting information.**

[Figure]

**Figure S10.** Aerosol mass spectrometry measurements of aerosol composition taken at the PEARL RidgeLab near Eureka showing only the organic and sulphate (SO₄) composition for GE3 **(a)** and GE6 **(c)** and the corresponding SMPS data for GE3 **(b)** and GE6 **(d)**. The sizes are mobility diameters measured by an SMPS, which are equal to the physical diameters under the assumption that the particles were spherical and contained no voids.

**RC3.9**

In addition, uncertainties on aerosol composition should be estimated and added to figure 7.

**The uncertainties are now described in the caption of the figure. We don't show the uncertainties in the figures because the overlapping error bars reduce the clarity of the figure.**

**RC3.10**

Finally, the authors conclude saying that "particles smaller than 100 nm in diameter are predominately organic with the organic-to-sulphate ratio increasing for smaller particle sizes", this is a general conclusion but is based on the analysis of only 2 events that does not provide any statistical basis for such kind of conclusion. For this reason the authors should look at the size resolved organic-to-sulfate ratio for all the events to check whether this statement is verified or not.

**We kindly refer the reviewer to our response to comment RC3.5.**

**RC3.11**

It would be useful also to compare the size resolved chemical composition with non event days.

**We agree that this would be extremely useful. Unfortunately, it was not possible to do such a comparison because the mass concentrations are too low.**

**Minor comments**

**RC3.12**

The authors often speak of particle nucleation, I would avoid using this terminology in the paper because there are no measurement for particles below 10 nm. For this reason there are no proof that nucleation is really happening at the measuring sites (in particular considering the small growth rates reported in this paper).

**We agree and the necessary changes have made throughout the manuscript.**

**RC3.13**

I would suggest to add a couple of sentences about iodine nucleation in the introduction. This has been proven to be a very effective mechanism in certain coastal regions [2] and a recent paper in the Arctic also showed evidence of this [3]. Later in the manuscript (chapter 3.2) the authors should also mention whether they see or not any evidence for iodine particles in their spectra.

**We kindly refer the reviewer to our response to comment RC1.1.**

**RC3.14**

In chapter 3.1.2 growth rates for 5 selected events are reported. However, there is no mention to the method used to estimate the growth, nor to the size range considered for the calculation. The authors should add this information to the manuscript that is really important in particular when comparing with results from other studies.

**We kindly refer the reviewer to our response to comment RC1.6.**

**RC3.15**

In chapter 3.2 a detailed analysis of fragments m/z 43 and 44 is provided, I wonder whether the authors can exclude any contamination from combustion or other sources (e.g. the generator) on fragment m/z 43.

**We kindly refer the reviewer to our response to comment RC2.5.**

**RC3.16**

Moreover, I agree with referee #1 in saying that figure 8 doesn't show any clear trend and the authors should reconsider their conclusions here.

**We kindly refer the reviewer to our response to comment RC1.8.**

**RC3.17**

Figure 3 shows a nice agreement between the PM1 as measured by the AMS and SMPS+OPC, however the authors should mention which density values were used to calculate the total mass for this comparison.

**The average density as a function of time was calculated from the AMS composition measurements and used to calculate the total mass for the comparison. For the entire AMS measurement period the average density was 1.31 g/cm³. The following has been added to Section 2.2:** *Applying the density calculated from the AMS data to the particle size distribution, a linear regression…*

**RC3.18**

Figure 4 reports the total particle number concentration at the 2 measurement sites, I would suggest to add a second box and whisker plot showing only the concentration of Aitken mode particles.

**We kindly refer the reviewer to our response to comment RC2.12.**

**NOTE: Modified text is highlighted in yellow (PLH)**

[revised manuscript text omitted]
 passed through a 6.35 mm OD and 4.72 mm ID copper tube that was 1.05 m long. For the OPC, the flow passed from the common aerosol inlet into a 12.7 mm OD copper tube with an ID of 9.40 mm and a length of 0.8 m, and then into a 6.35 mm OD copper tube with an ID of 4.72 mm and a length of 0.58 m, which was connected to the OPC by 6.35 mm OD conductive rubber tubing with an ID of 3.18 mm and a length of 0.04 m. Particle transmission efficiency to the SMPS has been calculated and the resulting transmission curve is shown in Figure S1 (von der Weiden et al. 2009). From the common aerosol inlet, the AMS flow passed first through 0.5 m of 25.4 mm OD and 22 mm ID stainless steel tubing and then through a 9.53 mm OD stainless steel tube with a length of 0.45 m before entering the AMS.

**Table S1.** List of all growth events observed near Eureka during the summers of 2015 and 2016 .

| Event Number | Date and Time (UTC) | | | Average temp. change from 10 to 600 m (°C) |
|---|---|---|---|---|
| | Start | | End | |
| GE 1 | 2015-07-27 18:00 | | 2015-07-28 03:00 | 2.00 |
| GE 3 | 2015-07-29 05:00 | | 2015-07-30 11:00 | 1.10 |
| GE 6 | 2015-08-02 04:00 | | 2015-08-03 05:00 | 0.20 |
| GE 9 | 2015-08-07 03:00 | | 2015-08-08 05:00 | 0.00 |
| GE 10 | 2015-08-07 21:00 | | 2015-08-09 02:00 | 1.23 |
| GE 15 | 2015-08-18 12:00 | | 2015-08-19 04:00 | 1.28 |
| GE 27 | 2016-06-15 00:00 | | 2016-06-17 18:00 | 1.20 |
| GE 28 | 2016-06-17 00:00 | | 2016-06-18 13:00 | 2.25 |
| GE 30 | 2016-06-25 20:00 | | 2016-06-27 14:00 | 1.30 |
| GE 32 | 2016-07-05 01:00 | | 2016-07-08 09:00 | 2.91 |
| GE 33 | 2016-07-08 07:00 | | 2016-07-09 01:00 | 0.00 |
| GE 34 | 2016-07-09 20:00 | | 2016-07-13 18:00 | 4.89 |
| GE 37 | 2016-07-20 18:00 | | 2016-07-22 03:00 | 1.80 |
| GE 38 | 2016-07-21 19:00 | | 2016-07-25 17:00 | 5.77 |
| GE 39 | 2016-07-27 02:00 | | 2016-07-27 17:00 | 0.47 |
| GE 43 | 2016-08-03 18:00 | | 2016-08-04 14:00 | 0.07 |
| GE 45 | 2016-08-05 03:00 | | 2016-08-05 23:00 | 0.00 |
| GE 46 | 2016-08-05 21:00 | | 2016-08-06 17:00 | 0.00 |
| GE 47 | 2016-08-13 19:00 | | 2016-08-17 12:00 | 0.46 |
| GE 48 | 2016-08-17 23:00 | | 2016-08-19 20:00 | 0.00 |
| GE 49 | 2016-08-20 23:00 | | 2016-08-22 15:00 | 0.00 |
| GE 51 | 2016-08-23 19:00 | | 2016-08-25 15:00 | 0.18 |
| GE 53 | 2016-08-26 19:00 | | 2016-08-27 20:00 | 2.27 |
| GE 54 | 2016-09-04 01:00 | | 2016-09-05 09:00 | 1.45 |
| GE 55 | 2016-09-05 09:00 | | 2016-09-06 01:00 | 4.30 |
| GE 56 | 2016-09-05 15:00 | | 2016-09-08 01:00 | 0.56 |
| GE 58 | 2016-09-10 12:00 | | 2016-09-12 16:00 | 1.28 |
| GE 59 | 2016-09-19 00:00 | | 2016-09-20 06:00 | 0.15 |

**Table S2.** List of selected periods of low and high particle concentrations observed near Eureka during the summers of 2015 and 2016 . The periods of high concentrations do not exhibit particle growth and are therefore distinct from the growth events. The SMPS measurements for the periods of low and high concentrations are shown in Figure S5 and S6, respectively.

| Non-Event (NE) | Time period (UTC) | | Average temp. change from 10 to 600 m (°C) |
|---|---|---|---|
| | Start | End | |
| NEA (low) | 2015-08-06   06:00 | 2015-08-07   02:00 | 0.00 |
| NEB (low) | 2015-08-24   11:00 | 2015-08-26   12:00 | 0.36 |
| NEC (low) | 2015-09-04   12:00 | 2015-09-05   18:00 | 0.60 |
| NED (low) | 2016-06-23   08:00 | 2016-06-24   15:00 | 0.13 |
| NEE (low) | 2016-06-30   12:00 | 2016-07-01   15:00 | 0.07 |
| NEF (low) | 2016-08-02   00:00 | 2016-08-02   12:00 | 0.50 |
| NEG (high) | 2015-08-10   07:00 | 2015-08-11   10:00 | 3.60 |
| NEH (high) | 2015-08-19   22:00 | 2015-08-24   10:00 | 1.98 |
| NEI (high) | 2015-08-29   07:00 | 2015-09-04   12:00 | 0.76 |
| NEJ (high) | 2016-07-13   19:00 | 2016-07-19   12:00 | 3.78 |
| NEK (high) | 2016-07-25   18:00 | 2016-07-26   03:00 | 2.90 |
| NEL (high) | 2016-07-30   00:00 | 2016-08-02   00:00 | 1.71 |

[Figure]

**Figure S1.** Inlet particle transmission efficiency for the AMS (a) and SMPS (b) at the PEARL RigdeLab. Note that the curves do not include the instrument transmission efficiencies for the AMS (Jayne et al. *Aerosol Sci. Technol.* **2000,** *33,* 49-70) or for the SMPS (Wiedensohler et al. *Atmos. Meas. Tech.* **2012,** *5,* 657-685).

**(a)**

[Figure]

**(b)**

[Figure]

**Figure S2.** Local map of PEARL RidgeLab, SAFIRE and Eureka weather station (a) and zoomed-out map of Alert and Eureka (b).

[Figure]

**Figure S3.** Scatter plots of the hourly particle number concentrations measured by the SMPS (300 − 487 nm) and the OPC (300 − 500 nm) near Eureka for the summers of 2015 (26 July to 26 September 2015) **(a)** and 2016 (16 June to 26 September 2016) **(b)**.

[Figure]

**Figure S4.** SMPS measurements of each growth event observed near Eureka during the summers of 2015 and 2016 as summarized in Table S1. The sizes are mobility diameters measured by an SMPS, which are equal to the physical diameters under the assumption that the particles were spherical and contained no voids.

[Figure]

**Figure S4.** SMPS measurements of each growth event observed near Eureka during the summers of 2015 and 2016 as summarized in Table S1 (continued). The sizes are mobility diameters measured by an SMPS, which are equal to the physical diameters under the assumption that the particles were spherical and contained no voids.

[Figure]

**Figure S4.** SMPS measurements of each growth event observed near Eureka during the summers of 2015 and 2016 as summarized in Table S1 (continued). The sizes are mobility diameters measured by an SMPS, which are equal to the physical diameters under the assumption that the particles were spherical and contained no voids.

[Figure]

**Figure S5.** SMPS measurements of selected periods with low particle concentrations observed near Eureka during the summers of 2015 and 2016 as summarized in Table S2. Note that the figures display an additional 2 hours before and after the analyzed period. The sizes are mobility diameters measured by an SMPS, which are equal to the physical diameters under the assumption that the particles were spherical and contained no voids.

[Figure]

**Figure S6.** SMPS measurements of selected periods with high particle concentrations and without growth observed near Eureka during the summers of 2015 and 2016 as summarized in Table S2. Note that the figures display an additional 2 hours before and after the analyzed period. The sizes are mobility diameters measured by an SMPS, which are equal to the physical diameters under the assumption that the particles were spherical and contained no voids.

[Figure]

**Figure S7.** Evaluation of the air mass history during the five selected growth events summarized in Table 1 and shown in Figure 5 of the main text. The back-trajectory and potential emissions sensitivity were calculated using FLEXPART. The left column corresponds to air masses arriving at Eureka and the right column corresponds to those arriving at Alert.

[Figure]

**Figure S7.** Evaluation of the air mass history during the five selected growth events summarized in Table 1 and shown in Figure 5 of the main text. The back-trajectory and potential emissions sensitivity were calculated using FLEXPART. The left column corresponds to air masses arriving at Eureka and the right column corresponds to those arriving at Alert (continued).

[Figure]

**Figure S8.** SMPS measurements of 4 growth events at Alert during the summers of 2015 and 2016 corresponding to the periods summarized in Table 1. The sizes are mobility diameters measured by an SMPS, which are equal to the physical diameters under the assumption that the particles were spherical and contained no voids.

**(a)**

[Figure]

**(b)**

[Figure]

**Figure S9.** Image of the ice coverage around Eureka during 25 June 2016 (a) and during 7 July 2016 (b) given by NASA Worldview.

[Figure]

**Figure S10.** Aerosol mass spectrometry measurements of aerosol composition taken at the PEARL RidgeLab near Eureka showing only the organic and sulphate ($SO_4$) composition for GE3 **(a)** and GE6 **(c)** and the corresponding SMPS data for GE3 **(b)** and GE6 **(d)**. The sizes are mobility diameters measured by an SMPS, which are equal to the physical diameters under the assumption that the particles were spherical and contained no voids.

[Figure]

**Figure S11.** AMS average ambient aerosol mass spectrum of GE 3 (a) and GE 6 (b) compared with the mass spectrum of MSA (c). The Ion Rate Fraction is the normalized Ion Rate (in Hz).

[Figure]

**Figure S12.** Organic aerosol fraction measured at m/z 43 and m/z 44 during GE 3 (a) and GE 6 (b) near Eureka.

[Figure]

**Figure S13.** Photos from the PEARL RidgeLab, at 610 m above sea level. The images correspond to: the first observed growth event for 2016 **(25 June)**, the first time open water is observed **(14 July)**, the last observed growth event for 2016 **(10 September)**, and the last image available for the year **(28 September)** due to poor visibility related to meteorological conditions as well as polar sunset.

[Figure]

**Figure S14.** Average by month of the downwelling shortwave radiation for the year 2016 at Eureka, as measured at the SAFIRE facility at 85 m above sea level. The standard deviation of the one-minute average fluxes for each month is indicated along with the average.

---

## Referee Report (RR1)

31st December 2018

The authors addressed most of my concerns with the second version of the manuscript and the other additional information. However, I still have some questions that I would like to see answered before publication.

**Major comments**

- **Aerosol composition during growth:** I would like to thank the authors for the additional information and figures provided to answer my previous concerns (RC 3.7 and 3.8). However I tend to disagree with their answer, in particular, the authors stated that "the organic size distribution is shifting to larger sizes from the beginning to the end of the growth event", whereas for my understanding there is no shift but just appearance/disappearance of different modes. Moreover, the Aitken mode that is growing contributes to a very small fraction of the total signal in the AMS and therefore is difficult to draw any conclusion about changes in the Aitken mode composition during the growth events. I'm particularly concerned about the last part of section 3.2 where the authors speculate about the change in SOA composition during GE3 and GE6. The average size-resolved measurement for m/z 43 and 44 is used to investigate the "amount of organic aerosol oxidation during growth events". However, the measurement is obtained averaging over the entire event, therefore it cannot provide any information about the evolution of particle composition over time, the fact that the ratio between the two organic fragments is different between the Aitken and the accumulation mode is not surprising and doesn't say anything about changes in the SOA composition during the growth. In addition, in figure 9 the authors plotted $f_{44}$ against $f_{43}$ to show that the SOA oxidation increases during the growth, I disagree with this conclusion because the mass concentration of these two organic fragments is dominated by particles that are larger than the Aitken mode under investigation. Just by looking at the aerosol size resolved chemical composition it is clear that a variation in the Aitken mode could not explain the variance in figure 9 that, in my opinion, is entirely due to variations in the concentration and/or composition of accumulation mode particles. The authors should provide more convincing argumentations to show that SOA oxidation is changing during the growth events, for example they could try to reproduce figure 9 excluding the contribution of larger particles, but I'm afraid the signal to noise ratio would be too low for this. In any case section 3.2 should be revised to address these comments.

- **Back-trajectory:** I would like the authors to link the results of the

back-trajectory analysis in section 3.1.3 with the observations reported in section 3.1.2 that shows that growth events are recorded only when a small temperature inversion is present. I think these two results are strictly connected and the authors should present them in a more coherent way. Moreover, I didn't find any indication about the height of the back-trajectories, I think this is an important information that should be provided in the paper.

**Minor comments**

- Page 8, lines 30-35: this sentence is overstated because it extrapolates the analysis of the results from two single events to many more. I agree with the authors in saying that there are evidences from different studies supporting the role of organics for aerosol growth in the Arctic but they are mostly based on the analysis of very few events. Therefore, I would not say that LVOC are responsible for the frequent particle growth events observed on Ellesmere island, there are not enough measurements to support this statement.

- Page 8, line 11: Is there any specific reason for choosing 10-30 nm as the size range to calculate the growth rate?

- Page 8, lines 14-15: this sentence is not very clear and should be rephrased with something like: In particular, the absence of an inversion below the PEARL RidgeLab would correspond to air masses measured at the site that are less photochemically aged and more influenced by local and possibly marine sources.

- Page 8, line 29: The authors showed here that the vertical structure of the atmosphere is similar during clean days and growth events. I would be curious to know if there are differences (e.g. airmass history) between these two cases that could explain why particle growth was detected only during certain days.

- Page 8, line 32: a weak inversion does not imply a low particle surface area as stated in this sentence,this causality link should be removed.

- Page 9, line 15-17: I don't think that the different size ranges used for the growth rate calculation could explain differences up to one order of magnitude in the average value. I would think that different condensable vapours concentration and/or different environmental condition (e.g. temperature, solar radiation, etc.) could play a much more important role in determining the aerosol growth rate.

- Page 10, line 22: from figure S11 it seems like the relative intensity of m/z 79 between the ambient and the lab spectra is comparable, thus I think this sentence should be deleted.

- Page 10, line 28: here it is said that m/z 79 is peaked at larger sizes and any MSA would be present in the accumulation mode but most of the AMS fragments are peaked at larger sizes because this is a mass-based instrument. Moreover, during GE6, m/z 79 shows a pattern below 100 nm that resembles the total organic particle size distribution. I would appreciate if the authors could modify the text to address this comment and include m/z 79 size distribution in the supplementary information.

- Page 13, line 6: I would not say that the measurements reported in this paper are in contrast with those reported by Giamarelou et al. because they were taken in a location that is thousands of kilometers away from Ellesmere island. I would not expect to have the same nucleation and growth processes across the two sites just because they are at a similar latitude. I would suggest to delete this comparison from the conclusion and move the description of Giamarelou et al. results to the introduction.

- Figure 5: it would be useful to add the center of the Aitken mode on top of the particle size distribution to guide the eyes and show the aerosol growth.

- Figure 8: I see the authors reasons for extrapolating the aerosol size distribution down to 10nm but I still think this is misleading and I would cut everything at 50 nm.

- Figure 9: the colors of the markers associated with the seven hours average are inverted in GE6 panel.

---

## Author Response (AR2)

**Key**

Black = Reviewer Comments, **Solid Blue = Responses,** *Italicized Blue = Modified Text*

We would like to thank the reviewers again for taking the time to review our manuscript and for their thoughtful comments. Their feedback has improved the manuscript. We have reproduced the reviewer comments in black text. For ease of review, our responses are given in blue text, while text that has been modified in the manuscript is quoted using blue italics. We would also like to point out that the numbering of the figures from the revised manuscript is used here in the responses.

**Referee 2 Comments**

**RC2.1**

RC1.9 response (new Fig S9 and in main text)

Please note that "Worldview" is the display/browser platform and that it should be noted in the caption that these are MODIS images.

**The caption and the main text have been updated.**

**RC2.2**

Page 1, Line 24

In the RC2.2 response, the authors note that the diameter range of "between 50 and 80 nm" corresponds to physical diameters, rather than vacuum aerodynamic diameter. Since this isn't stated until Section 2.2, please modify the text to make this clear here, for example "between 50 and 80 nm (physical diameter)".

**The text has been updated.**

**RC2.3**

RC2.3 response

Other Arctic growth event and aerosol size distribution literature that the authors might cite (and consider for interpretation of their data) include: Asmi et al (2011, ACP), Karl et al (2012, Tellus), Karl et al (2011, Tellus), Kolesar et al (2017, Atmos. Environ.).

**All of the citations have been added to the text in the introduction, except for Kolesar et al. which is now discussed in Section 3.1.2.**

**RC2.4**

Page 3, Line 10

Sipila et al. (2016, Nature) should also be cited here, as the authors show aerosol formation and growth due to iodic acid clusters at Station Nord.

**The text has been updated.**

**RC2.5**

RC2.9 response

For another comparison point, it might be helpful to note that Kolesar et al (2017, Atmos. Environ.) observed an average growth rate of 1.8 +/- 1.5 nm/h for spring-summer marine air masses at Barrow, AK.

**The text has been updated and the new text is copied below:**

*Moreover, Kolesar et al. (2017) observed an average growth rate of $1.8 \pm 1.5$ nm h$^{-1}$ for spring-summer marine air masses at Barrow, AK. In our study, growth rates ranged from $0.1 - 1.0$ nm h$^{-1}$ for the aerosols at the PEARL RidgeLab and at Alert, with an average rate of $0.5 \pm 0.3$ nm hr$^{-1}$ (Table 1). These values overlap with those reported in Collins et al. and even more similar to those in Nieminen et al. and Kolesar et al.*

**Major comments**

*Aerosol composition during growth:*

**RC3.1**

I would like to thank the authors for the additional information and figures provided to answer my previous concerns (RC 3.7 and 3.8). However I tend to disagree with their answer, in particular, the authors stated that "the organic size distribution is shifting to larger sizes from the beginning to the end of the growth event", whereas for my understanding there is no shift but just appearance/disappearance of different modes.

Moreover, the Aitken mode that is growing contributes to a very small fraction of the total signal in the AMS and therefore is difficult to draw any conclusion about changes in the Aitken mode composition during the growth events. I'm particularly concerned about the last part of section 3.2 where the authors speculate about the change in SOA composition during GE3 and GE6. The average size-resolved measurement for m/z 43 and 44 is used to investigate the "amount of organic aerosol oxidation during growth events". However, the measurement is obtained averaging over the entire event, therefore it cannot provide any information about the evolution of particle composition over time, the fact that the ratio between the two organic fragments is different between the Aitken and the accumulation mode is not surprising and doesn't say anything about changes in the SOA composition during the growth.

In addition, in figure 9 the authors plotted f44 against f43 to show that the SOA oxidation increases during the growth, I disagree with this conclusion because the mass concentration of these two organic fragments is dominated by particles that are larger than the Aitken mode under investigation. Just by looking at the aerosol size resolved chemical composition it is clear that a variation in the Aitken mode could not explain the variance in figure 9 that, in my opinion, is entirely due to variations in the concentration and/or composition of accumulation mode particles. The authors should provide more convincing argumentations to show that SOA oxidation is changing during the growth events, for example they could try to reproduce figure 9 excluding the contribution of larger particles, but I'm afraid the signal to noise ratio would be too low for this. In any case section 3.2 should be revised to address these comments.

**The reviewer raises valid concerns over our interpretation of the observations. Indeed, the attribution of changes in $f_{44}$ and $f_{43}$ to SOA oxidation over time is speculative. The last part of the 2$^{nd}$ last paragraph of this section has been revised to read as follows.**

*From Figures 8e and 8f, we speculate that the smaller and larger particles are reflective of SOA formed earlier and later during the two growth events, respectively. The greater fraction of the signal at m/z 44 in the accumulation mode relative to the Aitken mode would thus represent increased oxidation and greater production of carboxylic acids as*

*the events progressed. This would be consistent with the slight to moderate increase in $f_{44}$ observed in Figure 9 throughout both events. However, there was insufficient signal in our measurement to directly observe a change in $f_{44}$ in the Aitken mode aerosols to prove that oxidation actually increased. It is entirely possible that these observed differences were due to larger-scale processes that changed the overall aerosol population without SOA formation. Moreover, we emphasize that our results are for a very limited data set and further analysis of SOA composition during additional growth events using $f_{44}$ and $f_{43}$ would be necessary to confirm our observations and speculations.*

**Back-trajectory:**

**RC3.2**

I would like the authors to link the results of the back-trajectory analysis in section 3.1.3 with the observations reported in section 3.1.2 that shows that growth events are recorded only when a small temperature inversion is present. I think these two results are strictly connected and the authors should present them in a more coherent way.

**As described in the manuscript we have analyzed 17 periods in detail with respect to the inversion at PEARL. These periods are divided into 3 categories based on the aerosol size distribution: (1) growth events, (2) periods of low particle concentrations and (3) periods with a persistent accumulation mode and no growth. Back-trajectory calculations have been performed for all of these periods, and there is no clear change in the back-trajectories when comparing between the categorized events, and thus, when the temperature inversion is large or small. Therefore, it is difficult to make a strong connection between the discussion of the back-trajectories and that of the temperature profile. We have modified the text in Section 3.1.3 to include the new back-trajectory calculations.**

*Furthermore, we conducted a similar back-trajectory analysis for the six periods when particle concentrations were low and for the six periods with a persistent accumulation mode as described above in Section 3.1.2 and summarized in Table S2. The results are shown in Figures S10 and S11. There are no clear differences between the back-trajectories for the different types of periods and the growth events with almost all back-trajectories showing substantial potential emissions sensitivities over continental and marine regions mostly within the Arctic.*

**Furthermore, two new figures have been created in the supporting information.**

[Figure]

**Figure S10.** Evaluation of the air mass history during the low particle concentration events summarized in Table S2 and shown in Figure S5. The back-trajectory and potential emissions sensitivity were calculated using FLEXPART.

[Figure]

**Figure S11.** Evaluation of the air mass history during events with high particle concentrations and without growth summarized in Table S2 and shown in Figure S6. The back-trajectory and potential emissions sensitivity were calculated using FLEXPART.

**RC3.3**

Moreover, I didn't find any indication about the height of the back-trajectories, I think this is an important information that should be provided in the paper.

**The altitudes corresponding to the sites (PEARL and Alert) at which the air-tracer particles were released for the FLEXPART calculations are now given. We also note that the back-trajectories are plotted as potential emission sensitivity (PES), which represents the sensitivity of the receptor site to a specific source region, and therefore there is no height corresponding to the PES results (Seibert and Frank, 2004).**

Seibert, P. and Frank, A.: Source-receptor matrix calculation with a Lagrangian particle dispersion model in backward mode, Atmos. Chem. Phys., 4, 51-63, https://doi.org/10.5194/acp-4-51-2004, 2004.

**Minor comments**

**RC3.6**

Page 8, lines 30-35

This sentence is overstated because it extrapolates the analysis of the results from two single events to many more. I agree with the authors in saying that there are evidences from different studies supporting the role of organics for aerosol growth in the Arctic but they are mostly based on the analysis of very few events. Therefore, I would not say that LVOC are responsible for the frequent particle growth events observed on Ellesmere Island, there are not enough measurements to support this statement.

**The text has been updated in the introduction.**

*Taken together, these results provide important evidence that the condensation of lower volatility organic vapors on particle surfaces may be responsible, at least in part, for the particle growth events that are frequently measured at two sites on Ellesmere Island (e.g. approximately 20 events during summer 2016 at Eureka).*

**RC3.4**

Page 8, line 11

Is there any specific reason for choosing 10-30 nm as the size range to calculate the growth rate?

**We use this range to be consistent with the previous studies discussed in the manuscript where they also calculate the growth rate for the same diameters.**

**RC3.5**

Page 8, lines 14-15

This sentence is not very clear and should be rephrased with something like: In particular, the absence of an inversion below the PEARL RidgeLab would correspond to air masses measured at the site that are less photochemically aged and more influenced by local and possibly marine sources.

**The text has been updated.**

**RC3.6**

Page 8, line 29

The authors showed here that the vertical structure of the atmosphere is similar during clean days and growth events. I would be curious to know if there are differences (e.g. airmass history) between these two cases that could explain why particle growth was detected only during certain days.

**We kindly refer the reviewer to our response to comment RC3.2.**

**RC3.7**

Page 8, line 32

A weak inversion does not imply a low particle surface area as stated in this sentence, this causality link should be removed.

**The text has been updated as given below.**

*The results shown in Figure 7 imply that growth events occur at the PEARL RidgeLab when the inversion is weak because, firstly, those periods correlated with low particle surface area concentrations and corresponding condensation sink in the marine boundary layer air which allows particle nucleation to occur, and secondly, the site was possibly influenced by more recent surface emissions that were less photochemically aged compared to air aloft.*

**RC3.8**

Page 9, line 15-17

I don't think that the different size ranges used for the growth rate calculation could explain differences up to one order of magnitude in the average value. I would think that different condensable vapours concentration and/or different environmental condition (e.g. temperature, solar radiation, etc.) could play a much more important role in determining the aerosol growth rate.

**The relevant text has been updated.**

*It should be noted that the size range used for calculating growth rates in our work (10 – 30 nm) is slightly different from that of Collins et al. (4 – 20 nm) and Nieminen et al. (10 – 25 nm). However, it is more likely that different condensable vapour concentrations or different environmental conditions (e.g. temperature, solar radiation, etc.) led to the variations in the observed aerosol growth rates.*

**RC3.9**

Page 10, line 22

From figure S11 it seems like the relative intensity of m/z 79 between the ambient and the lab spectra is comparable, thus I think this sentence should be deleted.

**The figure for the MSA spectrum was incorrectly displayed so that all the m/z values were shifted by 1. This error only occurred in the MSA spectrum and not in the ambient spectra and it does not impact the discussion in the main text. The figure has been corrected and the m/z 79 value in the MSA spectrum is clearly much stronger than in the ambient spectra. (See next page)**

[Figure]

**Figure S13.** AMS average ambient aerosol mass spectrum of GE 3 (a) and GE 6 (b) compared with the mass spectrum of MSA (c). The Ion Rate Fraction is the normalized Ion Rate (in Hz).

RC3.10

Page 10, line 28

Here it is said that m/z 79 is peaked at larger sizes and any MSA would be present in the accumulation mode but most of the AMS fragments are peaked at larger sizes because this is a mass-based instrument. Moreover, during GE6, m/z 79 shows a pattern below 100 nm that resembles the total organic particle size distribution. I would appreciate if the authors could modify the text to address this comment and include m/z 79 size distribution in the supplementary information.

**The text has been updated as given below.**

*However, the size distribution of m/z 79 during GE 6 shows some signal below 100 nm, suggesting that MSA could be present in Aitken mode particles during at least some growth events (Figure S14).*

**Furthermore, a new figure has been created in the supporting information.**

[Figure]

**Figure S14.** Size distribution at m/z 79 for GE 6.

**RC3.11**

Page 13, line 6

I would not say that the measurements reported in this paper are in contrast with those reported by Giamarelou et al. because they were taken in a location that is thousands of kilometers away from Ellesmere Island. I would not expect to have the same nucleation and growth processes across the two sites just because they are at a similar latitude. I would suggest to delete this comparison from the conclusion and move the description of Giamarelou et al. results to the introduction.

**We agree with the review and thank them for this comment. We have updated the text accordingly.**

**RC3.12**

Figure 5

It would be useful to add the center of the Aitken mode on top of the particle size distribution to guide the eyes and show the aerosol growth.

**We have updated the figure as requested and as shown below.**

[Figure]

**Figure 5.** Five selected growth events near Eureka during the summers of 2015 and 2016. The grey dashed line indicates the start of each growth event and the black line indicates the Aitken mode diameter. The sizes are mobility diameters measured by an SMPS, which are equal to the physical diameters under the assumption that the particles were spherical and contained no voids.

**RC3.13**

Figure 8

I see the authors' reasons for extrapolating the aerosol size distribution down to 10nm but I still think this is misleading and I would cut everything at 50 nm.

**According to published work (Canagaratna et al. 2007), the AMS inlet shows 100% transmission efficiency for particles in an aerodynamic diameter range beginning at 70 nm and substantial transmission for particles in the 30 – 70 nm range. Therefore, there is still useful information to be gained from the AMS measurements below 50 nm, if the discussion of the results remains qualitative, as is the case in our manuscript. We have modified the size range in the figures to a lower limit of 20 nm, rather than 50 nm. We have chosen 20 nm because the AMS data is displayed with**

respect to physical diameter, and a 20 nm physical diameter corresponds approximately to a 30 nm aerodynamic diameter, given the average particle density.

Furthermore, we think this is an important point raised by the review and so we have added these considerations in Section 2.2 of the manuscript on the AMS measurements.

*The aerodynamic diameter was calibrated using polystyrene latex spheres at 80, 125, 240 and 300 nm. There are two important limitations to the size resolved AMS measurements reported here. Firstly, it should be noted that the extrapolation of the aerodynamic diameter calibration below 80 nm is not well constrained, so particle size data below this diameter should be considered qualitative rather than quantitative. Secondly, the AMS inlet has less than 100% transmission efficiency below aerodynamic diameters of 70 nm, although there is still substantial transmission of particles down to diameters of 30 nm.*

Lastly, we respectfully disagree with the description in the comment that we "extrapolated" the aerosol size distribution. The size distribution data was not extrapolated but rather the extrapolation was performed on the particle size calibration curve obtained using PSLs. This means that, while the sizes may not be accurate below 80 nm, the general trends in composition versus diameter are still valid. We have chosen to continue to display the AMS results between 20 – 50 nm because there is real scientific information contained in those results, and we think it is more important that the results be available to the scientific community, even if they are only qualitative.

Canagaratna, M., et al. Chemical and Microphysical Characterization of Ambient Aerosols with the Aerodyne Aerosol Mass Spectrometer, Mass Spectrom. Rev., 26, 185-222, https://doi.org/10.1002/mas.20115, 2007.

**RC3.14**

Figure 9

The colors of the markers associated with the seven hours average are inverted in GE6 panel.

We thank the review for catching this mistake and have corrected the figure.

[revised manuscript text omitted]
 passed through a 6.35 mm OD and 4.72 mm ID copper tube that was 1.05 m long. For the OPC, the flow passed from the common aerosol inlet into a 12.7 mm OD copper tube with an ID of 9.40 mm and a length of 0.8 m, and then into a 6.35 mm OD copper tube with an ID of 4.72 mm and a length of 0.58 m, which was connected to the OPC by 6.35 mm OD conductive rubber tubing with an ID of 3.18 mm and a length of 0.04 m. Particle transmission efficiency to the SMPS has been calculated and the resulting transmission curve is shown in Figure S1 (von der Weiden et al. 2009). From the common aerosol inlet, the AMS flow passed first through 0.5 m of 25.4 mm OD and 22 mm ID stainless steel tubing and then through a 9.53 mm OD stainless steel tube with a length of 0.45 m before entering the AMS.

**Table S1.** List of all growth events observed near Eureka during the summers of 2015 and 2016 .

| Event Number | Date and Time (UTC) | | | | Average temp. change from 10 to 600 m  (°C) |
|---|---|---|---|---|---|
| | Start | | End | | |
| GE 1 | 2015-07-27 | 18:00 | 2015-07-28 | 03:00 | 2.00 |
| GE 3 | 2015-07-29 | 05:00 | 2015-07-30 | 11:00 | 1.10 |
| GE 6 | 2015-08-02 | 04:00 | 2015-08-03 | 05:00 | 0.20 |
| GE 9 | 2015-08-07 | 03:00 | 2015-08-08 | 05:00 | 0.00 |
| GE 10 | 2015-08-07 | 21:00 | 2015-08-09 | 02:00 | 1.23 |
| GE 15 | 2015-08-18 | 12:00 | 2015-08-19 | 04:00 | 1.28 |
| GE 27 | 2016-06-15 | 00:00 | 2016-06-17 | 18:00 | 1.20 |
| GE 28 | 2016-06-17 | 00:00 | 2016-06-18 | 13:00 | 2.25 |
| GE 30 | 2016-06-25 | 20:00 | 2016-06-27 | 14:00 | 1.30 |
| GE 32 | 2016-07-05 | 01:00 | 2016-07-08 | 09:00 | 2.91 |
| GE 33 | 2016-07-08 | 07:00 | 2016-07-09 | 01:00 | 0.00 |
| GE 34 | 2016-07-09 | 20:00 | 2016-07-13 | 18:00 | 4.89 |
| GE 37 | 2016-07-20 | 18:00 | 2016-07-22 | 03:00 | 1.80 |
| GE 38 | 2016-07-21 | 19:00 | 2016-07-25 | 17:00 | 5.77 |
| GE 39 | 2016-07-27 | 02:00 | 2016-07-27 | 17:00 | 0.47 |
| GE 43 | 2016-08-03 | 18:00 | 2016-08-04 | 14:00 | 0.07 |
| GE 45 | 2016-08-05 | 03:00 | 2016-08-05 | 23:00 | 0.00 |
| GE 46 | 2016-08-05 | 21:00 | 2016-08-06 | 17:00 | 0.00 |
| GE 47 | 2016-08-13 | 19:00 | 2016-08-17 | 12:00 | 0.46 |
| GE 48 | 2016-08-17 | 23:00 | 2016-08-19 | 20:00 | 0.00 |
| GE 49 | 2016-08-20 | 23:00 | 2016-08-22 | 15:00 | 0.00 |
| GE 51 | 2016-08-23 | 19:00 | 2016-08-25 | 15:00 | 0.18 |
| GE 53 | 2016-08-26 | 19:00 | 2016-08-27 | 20:00 | 2.27 |
| GE 54 | 2016-09-04 | 01:00 | 2016-09-05 | 09:00 | 1.45 |
| GE 55 | 2016-09-05 | 09:00 | 2016-09-06 | 01:00 | 4.30 |
| GE 56 | 2016-09-05 | 15:00 | 2016-09-08 | 01:00 | 0.56 |
| GE 58 | 2016-09-10 | 12:00 | 2016-09-12 | 16:00 | 1.28 |
| GE 59 | 2016-09-19 | 00:00 | 2016-09-20 | 06:00 | 0.15 |

**Table S2.** List of selected periods of low and high particle concentrations observed near Eureka during the summers of 2015 and 2016 . The periods of high concentrations do not exhibit particle growth and are therefore distinct from the growth events. The SMPS measurements for the periods of low and high concentrations are shown in Figure S5 and S6, respectively.

| Non-Event (NE) | Time period (UTC) | | Average temp. change from 10 to 600 m (°C) |
|---|---|---|---|
| | Start | End | |
| NEA (low) | 2015-08-06  06:00 | 2015-08-07  02:00 | 0.00 |
| NEB (low) | 2015-08-24  11:00 | 2015-08-26  12:00 | 0.36 |
| NEC (low) | 2015-09-04  12:00 | 2015-09-05  18:00 | 0.60 |
| NED (low) | 2016-06-23  08:00 | 2016-06-24  15:00 | 0.13 |
| NEE (low) | 2016-06-30  12:00 | 2016-07-01  15:00 | 0.07 |
| NEF (low) | 2016-08-02  00:00 | 2016-08-02  12:00 | 0.50 |
| NEG (high) | 2015-08-10  07:00 | 2015-08-11  10:00 | 3.60 |
| NEH (high) | 2015-08-19  22:00 | 2015-08-24  10:00 | 1.98 |
| NEI (high) | 2015-08-29  07:00 | 2015-09-04  12:00 | 0.76 |
| NEJ (high) | 2016-07-13  19:00 | 2016-07-19  12:00 | 3.78 |
| NEK (high) | 2016-07-25  18:00 | 2016-07-26  03:00 | 2.90 |
| NEL (high) | 2016-07-30  00:00 | 2016-08-02  00:00 | 1.71 |

[Figure]

**Figure S1.** Inlet particle transmission efficiency for the AMS (a) and SMPS (b) at the PEARL RigdeLab. Note that the curves do not include the instrument transmission efficiencies for the AMS (Jayne et al. *Aerosol Sci. Technol.* **2000,** *33,* 49-70) or for the SMPS (Wiedensohler et al. *Atmos. Meas. Tech.* **2012,** *5,* 657-685).

**(a)**

[Figure]

**(b)**

[Figure]

**Figure S2.** Local map of PEARL RidgeLab, SAFIRE and Eureka weather station (a) and zoomed-out map of Alert and Eureka (b).

[Figure]

**Figure S3.** Scatter plots of the hourly particle number concentrations measured by the SMPS (300 – 487 nm) and the OPC (300 – 500 nm) near Eureka for the summers of 2015 (26 July to 26 September 2015) **(a)** and 2016 (16 June to 26 September 2016) **(b)**.

[Figure]

**Figure S4.** SMPS measurements of each growth event observed near Eureka during the summers of 2015 and 2016 as summarized in Table S1. The sizes are mobility diameters measured by an SMPS, which are equal to the physical diameters under the assumption that the particles were spherical and contained no voids.

[Figure]

**Figure S4.** SMPS measurements of each growth event observed near Eureka during the summers of 2015 and 2016 as summarized in Table S1 (continued). The sizes are mobility diameters measured by an SMPS, which are equal to the physical diameters under the assumption that the particles were spherical and contained no voids.

[Figure]

**Figure S4.** SMPS measurements of each growth event observed near Eureka during the summers of 2015 and 2016 as summarized in Table S1 (continued). The sizes are mobility diameters measured by an SMPS, which are equal to the physical diameters under the assumption that the particles were spherical and contained no voids.

[Figure]

**Figure S5.** SMPS measurements of selected periods with low particle concentrations observed near Eureka during the summers of 2015 and 2016 as summarized in Table S2. Note that the figures display an additional 2 hours before and after the analyzed period. The sizes are mobility diameters measured by an SMPS, which are equal to the physical diameters under the assumption that the particles were spherical and contained no voids.

[Figure]

**Figure S6.** SMPS measurements of selected periods with high particle concentrations and without growth observed near Eureka during the summers of 2015 and 2016 as summarized in Table S2. Note that the figures display an additional 2 hours before and after the analyzed period. The sizes are mobility diameters measured by an SMPS, which are equal to the physical diameters under the assumption that the particles were spherical and contained no voids.

[Figure]

**Figure S7.** Evaluation of the air mass history during the five selected growth events summarized in Table 1 and shown in Figure 5 of the main text. The back-trajectory and potential emissions sensitivity were calculated using FLEXPART. The left column corresponds to air masses arriving at Eureka and the right column corresponds to those arriving at Alert.

[Figure]

**Figure S7.** Evaluation of the air mass history during the five selected growth events summarized in Table 1 and shown in Figure 5 of the main text. The back-trajectory and potential emissions sensitivity were calculated using FLEXPART. The left column corresponds to air masses arriving at Eureka and the right column corresponds to those arriving at Alert (continued).

[Figure]

**Figure S8.** SMPS measurements of 4 growth events at Alert during the summers of 2015 and 2016 corresponding to the periods summarized in Table 1. The sizes are mobility diameters measured by an SMPS, which are equal to the physical diameters under the assumption that the particles were spherical and contained no voids.

**(a)**

[Figure]

**(b)**

[Figure]

**Figure S9.** Image of the ice coverage around Eureka during 25 June 2016 (a) and during 7 July 2016 (b) as shown in MODIS imagery accessed by NASA Worldview.

[Figure]

**Figure S10.** Evaluation of the air mass history during the low particle concentration events summarized in Table S2 and shown in Figure S5. The back-trajectory and potential emissions sensitivity were calculated using FLEXPART.

[Figure]

**Figure S11.** Evaluation of the air mass history during events with high particle concentrations and without growth summarized in Table S2 and shown in Figure S6. The back-trajectory and potential emissions sensitivity were calculated using FLEXPART.

[Figure]

**Figure S12.** Aerosol mass spectrometry measurements of aerosol composition taken at the PEARL RidgeLab near Eureka showing only the organic and sulphate ($SO_4$) composition for GE3 **(a)** and GE6 **(c)** and the corresponding SMPS data for GE3 **(b)** and GE6 **(d)**. The sizes are mobility diameters measured by an SMPS, which are equal to the physical diameters under the assumption that the particles were spherical and contained no voids.

[Figure]

**Figure S13.** AMS average ambient aerosol mass spectrum of GE 3 (a) and GE 6 (b) compared with the mass spectrum of MSA (c). The Ion Rate Fraction is the normalized Ion Rate (in Hz).

[Figure]

**Figure S14. Size distribution at m/z 79 for GE 6.**

[Figure]

**Figure S15.** Organic aerosol fraction measured at m/z 43 and m/z 44 during GE 3 (a) and GE 6 (b) near Eureka.

[Figure]

**Figure S16.** Photos from the PEARL RidgeLab, at 610 m above sea level. The images correspond to: the first observed growth event for 2016 **(25 June)**, the first time open water is observed **(14 July)**, the last observed growth event for 2016 **(10 September)**, and the last image available for the year **(28 September)** due to poor visibility related to meteorological conditions as well as polar sunset.

[Figure]

**Figure S17.** Average by month of the downwelling shortwave radiation for the year 2016 at Eureka, as measured at the SAFIRE facility at 85 m above sea level. The standard deviation of the one-minute average fluxes for each month is indicated along with the average.